# Anti-Proliferative and Anti-Metastatic Effects of Ethanol Extract from *Cajanus cajan* (L.) Millsp. Roots and its Sub-Fractions in Oral Squamous Cell Carcinoma

**Thuy-Lan Thi Vo** [1], **Shu-Er Yang** [2], **Liang-Gie Huang** [3], **Po-Hsien Li** [4], **Chien-Lin Chen** [1] **and Tuzz-Ying Song** [1,*]

1 Department of Medicinal Botanicals and Foods on Health Applications, Da-Yeh University, Changhua 515, Taiwan
2 Department of Beauty Science and Graduate Institute of Beauty Science Technology, Chienkuo Technology University, Changhua 500, Taiwan
3 Department of Stomatology, Taichung Veterans General Hospital, Taichung 407, Taiwan
4 Department of Food and Nutrition, Providence University, Taichung 43301, Taiwan
* Correspondence: song77@mail.dyu.edu.tw; Tel.: +886-4-8511-888 (ext. 2282); Fax: +886-4-8511-320

**Abstract:** *Cajanus cajan* (L.) Millsp., known as pigeon pea, *C. cajan* (L.) Millsp. roots (CR) contain daidzein, genistein, and cajanol which have numerous health benefits. The aim of this study was to investigate the anti-proliferative and anti-metastatic effects of sub-fractions (EECRpw, EECRp25, EECRp50, EECRp70, and EECRp95) containing 95% ethanol extract from CR (EECR95) in oral squamous cell carcinoma cells (SCC25). We found that the sub-fraction (EECRp70) significantly inhibited cell proliferation, and down-regulated secretion of matrix metalloproteinase-2 and vascular endothelial growth factor-2 in a dose-dependent manner, and the mechanisms were related to down-regulated iNOS/COX-2/NF-kB signaling pathways. Moreover, EECRp70 in combination with a cancer chemotherapeutic agent (cisplatin or taxol) also blocked cell proliferation, arrested cell cycle in subG1 phase, inhibited intracellular production of reactive oxygen species, and reduced migration and invasion of SCC25 cells. Furthermore, our results demonstrate that EECRp70 inhibits proliferation and metastatic, which is related to the relatively high uptake of genistein of EECRp70 (2.71%) in SCC25 cells for 12 h. In addition, EECRp70 contains abundant flavonoids such as genistein and cajanol. Thus, we suggest that the study supports the anecdotal use of CR in herbal medicine and functional food.

**Keywords:** *Cajanus cajan* (L.) Millsp. roots; fractionation; anti-proliferation; synergism; anti-metastasis; and SCC25 cells

## 1. Introduction

Oral squamous cell carcinoma (OSCC) is the most common malignancy of the oral cavity (e.g., lip, oral cavity, oropharynx, and hypopharynx), with a high mortality rate in Taiwan and worldwide [1–3]. Many factors including chewing of betel quid, cigarette smoking, alcohol, ultraviolet radiation, human papillomavirus (HPV), candida infections, nutritional deficiencies, and genetic predisposition have also been risk factors for OSCC [4–7]. Common treatments include chemical drug therapy, radiation, and surgery; however, the survival rate of OSCC is lower and the side effects are significant [4]. Factors such as tumor grade, ability to evade apoptosis, increased angiogenesis, invasion, and metastasis affect the survival rate in OSCC [8,9]. Therefore, existing protocols for prevention, diagnosis, treatment, and support need to be further improved.

*C. cajan* (L.) Millsp, known as pigeon pea, is a highly valued medicinal plant and is cultivated in the tropics and semi-arid tropics. It belongs to the Fabaceae (or Leguminosae) family and has a high content of good quality proteins, dietary fiber, phenolic compounds,

and a low content of saturated fat [10,11]. Leaves, and roots of *C. cajan* contain high levels of polyphenols such as luteolin and apigenin [12], isoflavonoids such as daidzein, genistein, genistin, cajanol, biochanin A, cajaninstilbene, pinostrobin, coumarin, vitexin, and orientin [13–18]. Flavonoids have been shown to be cytotoxic to many human cancer cells, including oral cancer cells, breast cancer cells, colorectal cells, prostate carcinoma cells, and cervical carcinoma cells [19–23]. However, an article reported the synergistic effects of flavonoids in combination with chemotherapy agents on the proliferation and apoptosis in human cancer cell lines [22].

In vivo and in vitro studies have reported the biological activities of flavonoid compounds (cajanol, daidzein, and genistein) from the roots of *C. cajan* (CR), mainly related to anti-diabetic, antioxidant, anti-hyperglycemic, anti-hyperlipidemic, hypocholesterolemic, anti-inflammatory, and anti-bacterial activities [24–30]. As for anti-cancer properties, extracts from CR were reported to interfere with the proliferation of MCF-7 human breast cancer cells [15]. Cajanol, a novel isoflavone from CR, arrested the cell cycle in the G2/M phase and induced apoptosis via a reactive oxygen species (ROS)-mediated mitochondria-dependent pathway of MCF-7 human breast cancer cells. However, the underlying cell line, mechanism, and bioactive compounds remain to be investigated. Therefore, the aim of this study was to investigate the anti-proliferative and anti-metastatic effects of 95% ethanol extract from *C. cajan* roots (EECR95) and its sub-fractions in OSCC, and its possible mechanisms. We also investigated the synergistic effect of the combination of EECRp70 and chemotherapeutic agents (cisplatin or taxol) in inhibiting OSCC cell proliferation to identify whether EECR could be used as an adjuvant for cancer treatment to reduce the side effects of chemotherapeutic agents.

In the present study, we investigated the anti-cancer activity of CR against oral cell squamous (SCC25) in vitro. For this purpose, we first separated the 95% ethanol extract from CR (EECR95) to obtain five sub-fractions of (EECRpw, EECRp25, EECRp50, EECRp70, and EECRp95) and analyzed their potential bioactive compounds. We selected the most cytotoxic sub-fraction for further evaluation of the anti-proliferative effect, cell cycle regulation, ROS production, protein expression (NF-kB/iNOS/COX-2), and anti-metastatic activities (secretion of MMP-9 and VEGF-2, and migrated, invaded cells). We also studied the synergy between the potential sub-fraction and taxol or cisplatin (Figure 1).

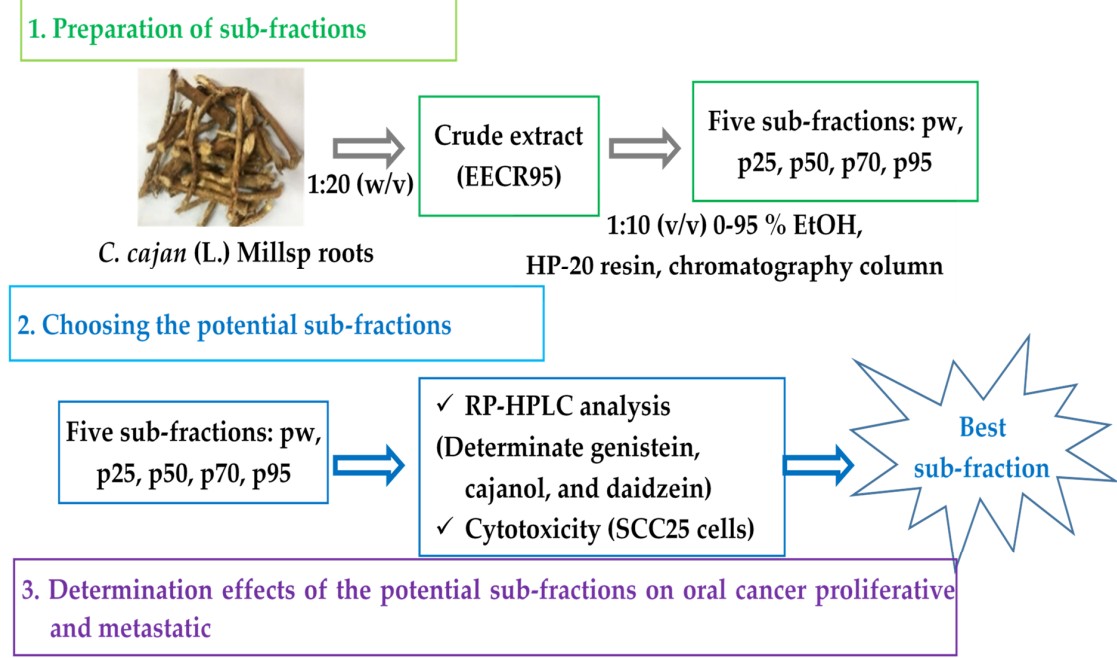

**Figure 1.** *Cont.*

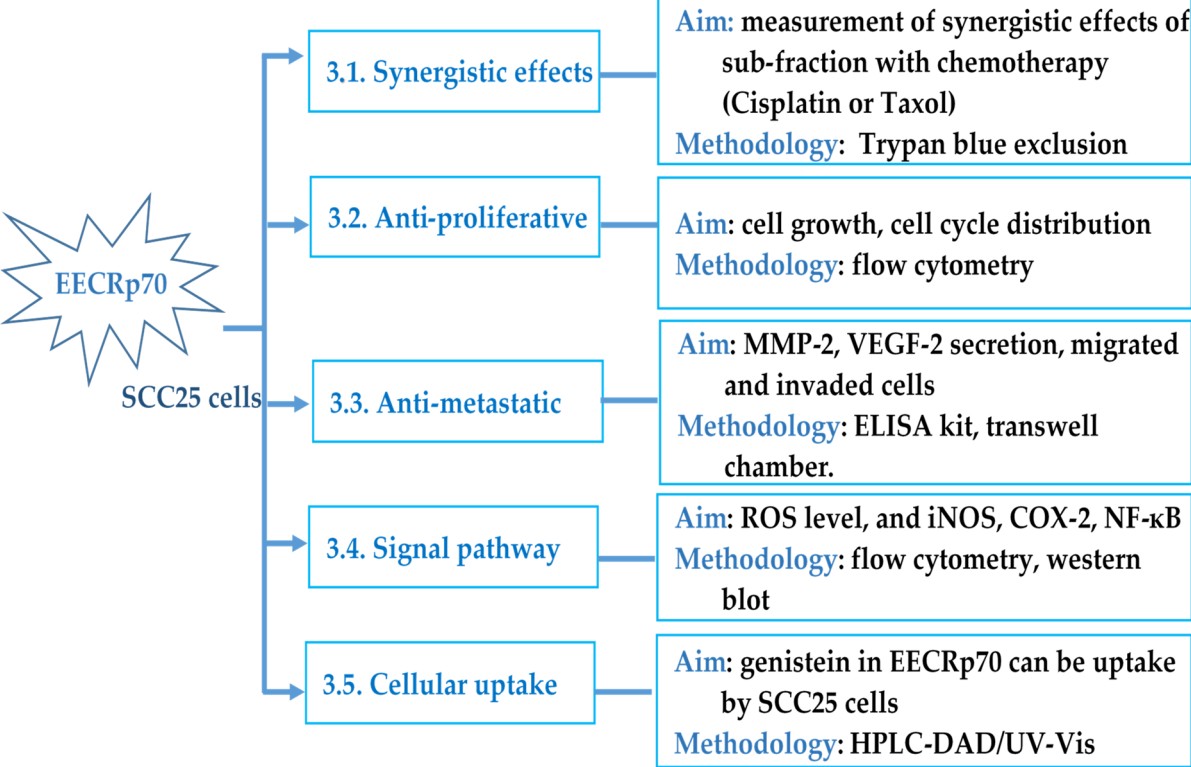

**Figure 1.** Flowchart of the best sub-fractions of the 95% ethanol extract from the roots of *C. cajan* (L.) Millsp. (EECR95) on anti-proliferative and anti-metastatic effects on oral squamous cell carcinoma (SCC25 cells).

## 2. Materials and Methods

### 2.1. Materials

Human oral squamous cell carcinoma (SCC25) was purchased from BCRC (Hsinchu, Taiwan). Cisplatin, paclitaxel (taxol), genistein, dimethyl sulfoxide (DMSO), PI/RNase staining solution, and trypan blue were purchased from Sigma-Aldrich Corp (St. Louis, MO, USA). Dulbecco's modified Eagle medium/F12 (DMEM/F12), fetal bovine serum (FBS), sodium pyruvate, hydrocortisone, sodium bicarbonate, trypsin-EDTA, and Giemsa were purchased from Life Technologies (Carlsbad, CA, USA). Primary antibodies for cyclooxygenase-2 (COX-2), inducible nitric oxide synthase (iNOS), and nuclear factor kappa-light-chain-enhancer of activated B cells (NF-κB) and secondary antibodies were purchased from Santa Cruz Biotechnology, Inc. (Santa Cruz, CA, USA). Extracellular matrix (ECM) was purchased from Engelbreth Holm Swarm murine sarcoma (Sigma-Aldrich Corp).

### 2.2. Preparation of Sub-Fraction from C. cajan

Dried CR was collected from Taitung District Agricultural Research and Extension Station, Council of Agriculture, Executive Yuan. The powdered roots (1000 g) were extracted with 95% ethanol at a ratio of 1:20 by soaking at room temperature for 72 h, filtering and concentrating under reduced pressure (10-fold), and freeze drying to obtain EECR95, with a yield about 28.90 (g/kg). HP-20 macroporous resin of the chromatography column was used to fractionate the EECR95 with a 1:10 (*w/v*) ratio of distilled water: 25%, 50%, 70%, and 95% ethanol at room temperature. Subsequently, the sub-fractions were concentrated (10-fold) to give EECRpw, EECRp25, EECRp50, EECRp70, and EECRp95, and the yields were 10.79, 2.93, 4.03, 2.17, and 5.82 g/kg, respectively. EECR95 and its sub-fractions were stored at 4 °C for the following analysis.

### 2.3. Determination of Genistein, Cajanol, and Daidzein

The analysis was carried out using an Agilent 1200 reversed-phase HPLC system (Hitachi, Chromaster 5430 Diode Array Detector). A HIQ Sil C18W reversed-phase column (250 mm × 4.6 mm i.e., 5 μm). The composition of polyphenols (genistein, cajanol, and daidzein) was measured according to the method presented in Vo et al. [14]. The results were expressed in mg/g dry weight.

### 2.4. Cell Culture and Cell Anti-Proliferative Assay

SCC25 cells were cultured in DMEM/F12, supplemented with 5% FBS, 1.2 g/L sodium bicarbonate, 1% sodium pyruvate, 40 μg/L hydrocortisone at 37 °C, and 5% $CO_2$. The medium was changed every two days. Cells were seeded at a density of $1 \times 10^5$ cells/well into a 12-well plate 24 h before treatment. After incubation, cells were washed twice with PBS × 1, and lysed with trypsin EDTA 0.25% (Gibco, CAD). The morphological changes were observed with a microscope (BestScope-7000B, Beijing, China). The number of viable cells was determined at 24, 48, and 72 h incubation by the trypan blue exclusion method using a hemocytometer.

The potential synergistic or antagonistic effects of EECRp70 in combination with cisplatin or taxol are consistent with the reports of Song et al. and Meyer et al. [31,32]. The formula for percent inhibition is as follows:

$$\frac{\text{control} - (\text{EECRp70} + \text{chemotherapeuric})}{(\text{control} - \text{EECRp70}) + (\text{control} - \text{chemotherapeutic drug})}$$

According to this formula, a value $\geq 1.0$ is synergistic; a value between $0.5 \leq$ and $\leq 1.0$ is additive; and a value $\leq 0.5$ is antagonistic.

### 2.5. MMP-2 and VEGF-2 Release Assay

To determine the effects of EECRp70 on MMP-2 and VEGF-2 levels, SCC25 was treated with various concentrations of EECRp70 (0–100 μg/mL) and 25 μM genistein for 24 h. The medium was then aspirated from the flasks and centrifuged at $500 \times g$ (10 min) to remove the cells from the medium. The levels of MMP-2 and VEGF-2 released into the incubation medium were determined using an ELISA kit [abcam, Taiwan] as described in Song et al. [31].

### 2.6. Cell Cycle Analysis via Flow Cytometry

The cell cycle was assessed by propidium iodide (PI)-DNA staining in a flow cytometry assay, this method was described in Gong et al. [33]. Cells were cultured, incubated with 25 μM genistein, 10 nM taxol, 25 μg/mL EECRp70, and 25 μg/mL + 10 nM taxol for 24 h. The harvest cells were collected and lysed with trypsin, and then centrifuged at 900 rpm for 10 min at 4 °C. The cells were fixed with 70% EtOH (in PBS × 1), and left to stand on ice for 30 min. Then cells were washed with PBS × 1. Then, the cells were incubated with PI/RNase solution for 30 min in the dark at 37 °C. The cells were transferred to a flow cytometer tube to measure cell cycle progression by flow cytometry (BD FACSCantoTM II, Franklin Lakes, NJ 07417, USA).

### 2.7. Measurement of Intracellular Reactive oxygen Species

The intracellular level ROS was measured using the fluorescent probe $2', 7'$- dichlorodi-hydrofluorescein diacetate ($H_2$DCFDA), as previously described in Lautraite et al. [34]. DCFDA readily diffuses through the cell membrane and is enzymatically hydrolyzed by intercellular esterase to form nonfluorescent DCFH, which is then rapidly oxidized to form highly fluorescent DCF in the presence of ROS. SCC25 cells ($1 \times 10^6$ cells/mL) were seeded in a 10 cm dish and incubated with 25 μM genistein, 10 nM taxol, 25 μg/mL EECRp70, and 25 μg/mL + 10 nM taxol for 24 h. The cells were then treated with 30 μM $H_2$DCFDA at 37 °C for 1 h in dark. The staining was removed by centrifugation at 4 °C, $900 \times g$ for 10 min.

The cell pellet was collected, and 0.5 mL of 1 × PBS was added, mixed, and meshed before transferring to a flow cytometer tube for measuring the ROS level using a flow cytometer (BD FACSCantoTM II, Franklin Lakes, NJ 07417, USA).

### 2.8. Cell Migration and Invasion Assay

Cell migration and invasion were studied in a tanswell chamber (BIOFIL, Guangzhou, China) using 6.5 mm polycarbonate filters with a pore size of eight μm according to the method described by Song and Repesh et al. with minor modifications [31,35]. The difference between the cell migration and invasion assays is that each upper transwell for the invasion assay was pre-coated with 100 μL of a 1:20 diluted Matrigel in cold DMEM/F12. SCC25 cells were pre-incubated with 0–100 μg/mL EECRp70, 25 μM genistein, 10 nM taxol, and 25 μg/mL + 10 nM taxol for 24 h, cell density was adjusted to $1 \times 10^5$ cells/300 μL for migration and $5 \times 10^4$ cells/300 μL for invasion in serum-free DMEM/F12 and the cell were added to the upper chamber. The lower chamber was filled with 600 μL of DMEM/F12, incubated at 37 °C, and 5% $CO_2$ for 24 h. The cells in the upper chamber were fixed with formaldehyde (3.7% in PBS), permeabilized with methanol, stained with Giemsa, wiped completely with a cotton swab, and photographed under a microscope (200×). For each replicate, cancer cells in six randomly selected fields were determined by counting and averaging the number.

### 2.9. Western Blot Analysis for the Protein Expressions

SCC25 cells were pretreated with EECRp70 (0–100 μg/mL) and 25 μM genistein for 24 h incubation. Western blot analysis was performed to detect iNOS, COX-2, NF-kB, β-actin, and lamin b protein on SCC25 cells, as described in [14].

### 2.10. Cellular Uptake

SCC25 cells ($1 \times 10^6$ cells/mL) were seeded in 10 cm dish and treated with 50 μg/mL (contained 5.01 and 1.90 μg/mL of genistein and cajanol) EECRp70 or 25 μM genistein and cajanol (6.80 and 7.90 μg/mL, respectively) and incubated for 3, 6, 12, and 24 h at 37 °C. Cells were collected, lysed, and centrifuged at 12,000× *g* for 30 min. A portion of the supernatant was applied to an HPLC-DAD/UV–Vis (Hitachi, Chromaster 5430 DAD), as described in Vo et al. [14]. Cellular uptake was expressed as ng/$10^6$ cells.

### 2.11. Statistical Analyses

Data are expressed as means ± SD, n = 3. Statistical analyzes were performed using Sigma blot 12.5 (Systat Software Inc. (SSI), San Jose, CA, USA) one-way ANOVA followed by Duncan's multiple range test ($p < 0.05$).

## 3. Results

### 3.1. The Contents of Flavonoids of EECR95 and Its Sub-Fractions

The HPLC profile of the 10 standard flavonoid compounds, the CR extract (EECR95) and sub-fractions of ethanol extract (EECRpw, EECRp25–p95: 25–95% ethanol sub-fraction of EECR95) from CR was determined at 290 nm using the HPLC-DAD-UV/Vis system. As the result shown in Figure 2, three kinds of flavonoids (genistein, daidzein and cajanol) were detected in EECR95; no flavonoid peak was detected in EECRpw; genistein peak was detected in EECRp25–95; daidzein peak was detected in EECRp50; and cajanol peak was detected in EECRp70.

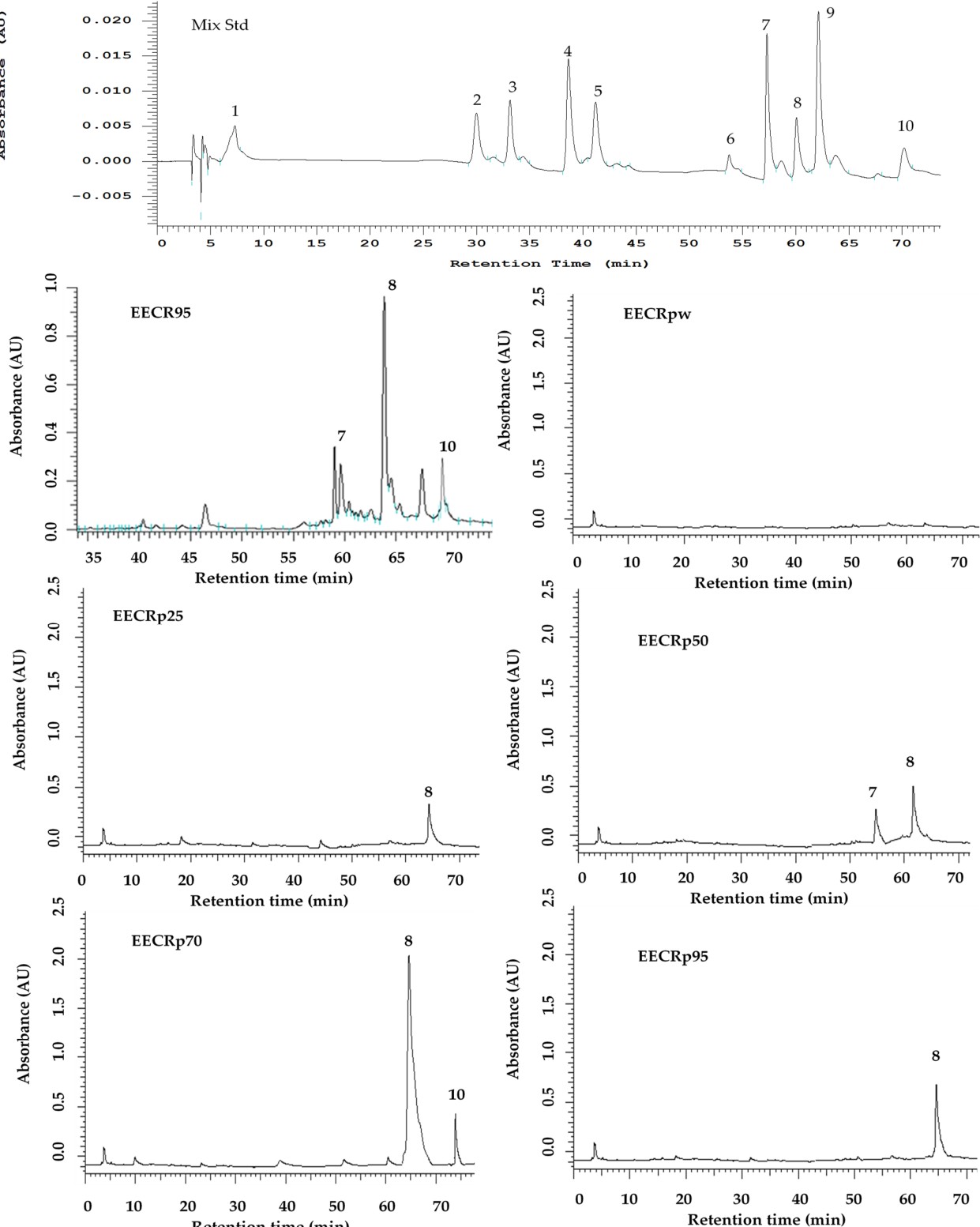

**Figure 2.** High performance liquid chromatography chromatograms of flavonoids composition of EECR95 (95% ethanol extract from *C. cajan* (L) Millsp. roots), EECRpw (water eluent of EECR95) and EECRp25-p95 (25–95% ethanol eluent of EECR95). Peaks 1–10 are as follows: (1) gallic acid; (2) Vannilic acid; (3) Syringic acid; (4) Courmaric acid; (5) Ferullic acid; (6) Rutin; (7) daidzein (8) genistein; (9) quercetin; and (10) cajanol.

Table 1 indicated that genistein, cajanol and daidzein were the major flavonoids in EECR95, and the contents were 19.47-, 3.72- and 2.68- mg/g, respectively. Genistein content (101.97 mg/g DW) was the highest followed by cajanol (37.74 mg/g DW) in EECRp70, and daidzein was detected only in EECRp50. In addition, the contents of cajanol, and genistein in EECRp70 were 10.1-, and 5.2-folds of EECR95, respectively.

**Table 1.** The contents of flavonoids of EECR95 and its sub-fractions.

| Sub-Fractions | Flavonoids (mg/g) | | |
| --- | --- | --- | --- |
| | Genistein | Cajanol | Daidzein |
| EECR95 | 19.47 ± 1.92 [b] | 3.72 ± 0.11 [a] | 2.68 ± 0.14 [a] |
| EECRpw | ND | ND | ND |
| EECRp25 | 2.57 ± 0.42 [a] | ND | ND |
| EECRp50 | 10.97 ± 3.02 [b] | ND | 4.67 ± 1.20 [a] |
| EECRp70 | 101.97 ± 1.28 [c] | 37.74 ± 0.50 [b] | ND |
| EECRp95 | 14.34 ± 0.21 [b] | ND | ND |

EECR95: extraction of *C. cajan* (L) Millsp. roots in 95% ethanol, EECRpw: sub-fraction of EECR95 in water, EECRp25–p95: 25–95% ethanol sub-fraction of EECR95. Value (means ± SD, n = 3 for the test groups) in each column that do not have the same superscript are significantly different ($p < 0.05$). ND: not detectable.

### 3.2. Cytotoxicity Effects of EECR95 and Its Sub-Fractions in SCC-25 Cells

Figure 3 shows that except EECRpw, EECR95, and its sub-fractions had a dose-dependent cytotoxic inhibitory effect on SCC25 cells. At a concentration of 100 µg/mL, the cell viability of EECRpw, EECRp25, EECRp50, EECRp70, EECRp95 was 95.53%, 94.46%, 76.69%, 65.23%, 60.64%, and 61.02% for 24 h incubation, respectively. After SCC25 cells were treated with EECRp70 for 24 h, cell viability at low concentration (20 µg/mL) ($p < 0.05$) (23.5%) was significantly lower than that of EECR95 and other sub-fractions. To avoid cytotoxic effects caused by high concentrations, we chose EECRp70 at a concentration of 10–100 µg/mL for the following experiments.

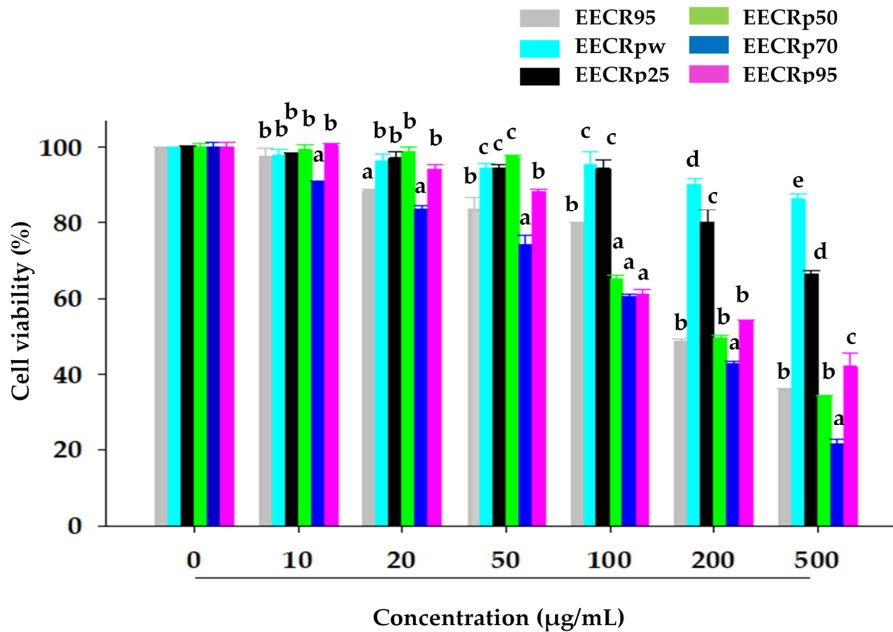

**Figure 3.** Effects of EECR95 and its sub-fractions on cell viability of SCC25 cells during a 24 h incubation. Cells were treated with (0–500 µg/mL) sub-fraction for 24 h incubation. After treatment, cell viability was measured by MTT assay. EECR95: 95% ethanol of *C. cajan* roots, EECRpw: sub-fraction of EECR95 in water, EECRp25–p95: 25–95% ethanol sub-fraction of EECR95. Values (mean ± SD, n = 3) that do not share a common letter are significantly different ($p < 0.05$).

### 3.3. Anti-Proliferation Effect of EECRp70 in SCC25 Cells

As shown in Figure 4, EECRp70 inhibited the anti-proliferative effect of SCC25 cells in a dose-dependent manner, and the $IC_{50}$ values at 24, 48, and 72 h were 107.53, 87.46, and 74.99 µg/mL, respectively. In addition, we found that the inhibitory effect of 100 µg/mL EECRp70 (about 10.2 µg/mL genistein) was slightly higher than that of 25 µM for genistein (6.8 µg/mL genistein) ($p < 0.05$).

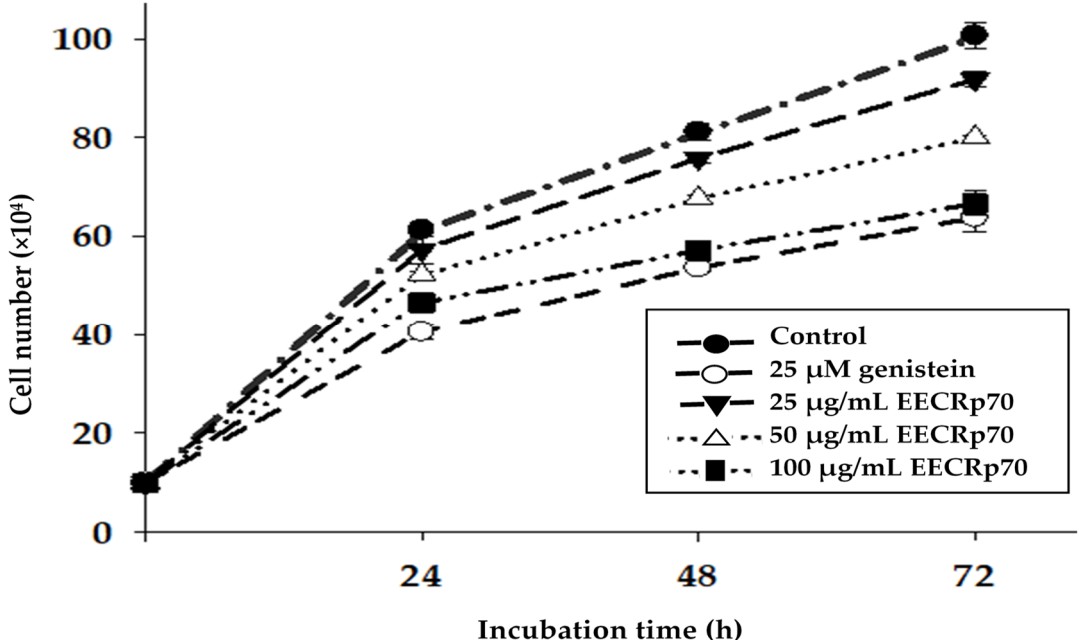

**Figure 4.** Effects of EECRp70 on anti-proliferation of SCC25 cells. Cells were treated with (0–100 µg/mL) EECRp70 for 24, 48, and 72 h. The number of viable cells was determined by the trypan blue exclusion method using a hemocytometer: 25 µM genistein (6.8 µg/mL genistein) and 100 µg/mL EECRp70 (10.2 µg/mL genistein). Results were expressed as (mean ± SD, n = 3, $p < 0.05$).

### 3.4. Effect of EECRp70 on MMP-2 and VEGF-2 Activity of SCC25 Cells

As shown in Figure 5, treatment of SSC25 cells with EECRp70 for 24 h resulted in a dose-dependent inhibition of MMP-2 and VEGF-2 levels ($p < 0.05$). The percentage inhibition of MMP-2 levels by 25, 50, and 100 µg/mL EECRp70 was 18.2%, 46.2%, and 74.3%, respectively ($IC_{50}$ = 63.36 µg/mL) (Figure 5A), and the inhibition of the VEGF level was 18.2%, 24.7%, and 82.2%, respectively ($IC_{50}$ = 76.88 µg/mL) on (Figure 5B). In addition, the inhibitory effects of 50 µg/mL of EECRp70 on MMP-2 and VEGF-2 levels were approximately the same as those of 25 µM for genistein ($p < 0.05$).

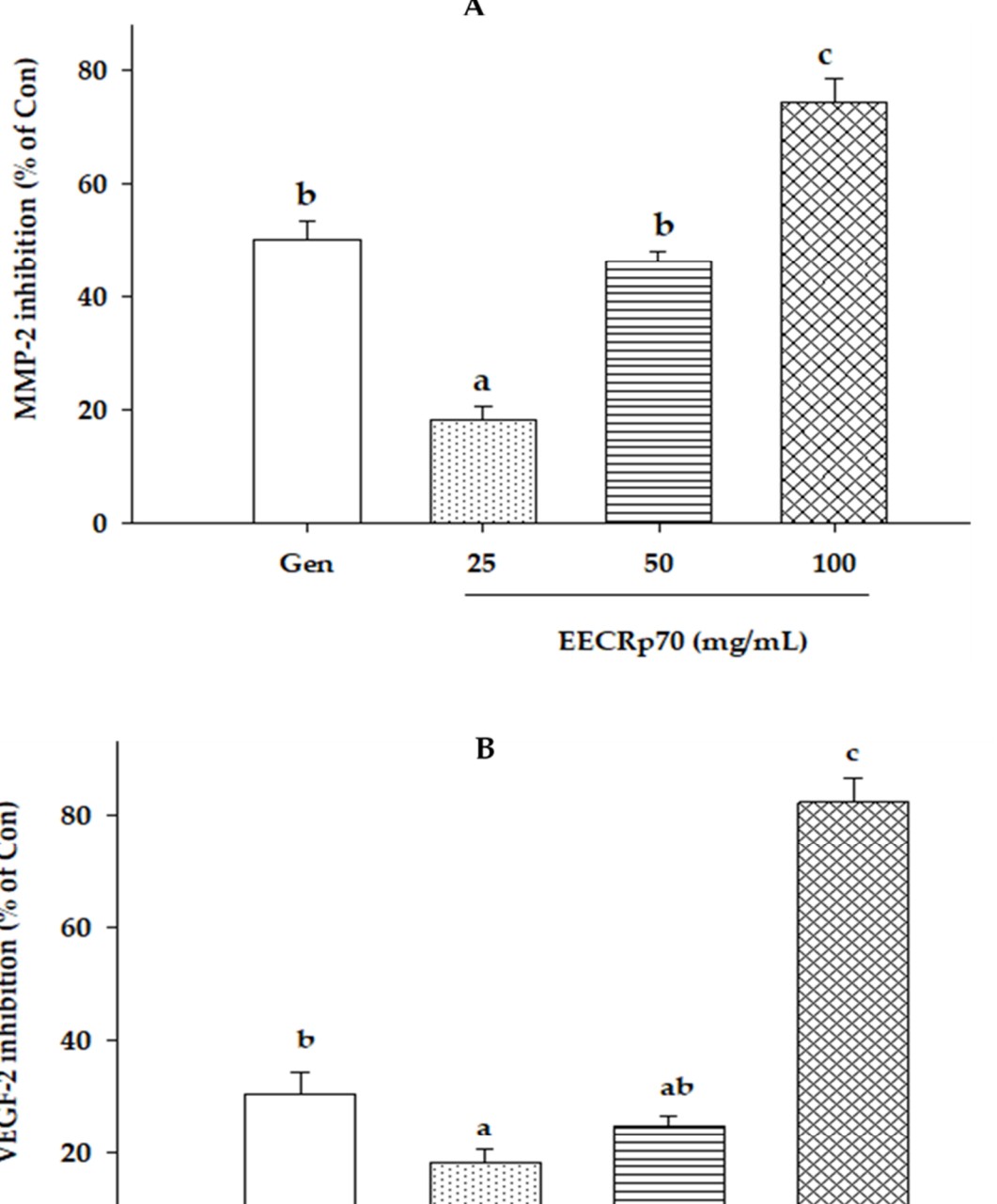

**Figure 5.** Effect of EECRp70 on the levels of (**A**) matrix metalloproteinase-2 (MMP-2) and (**B**) vascular endothelial growth factor-2 (VEGF-2) in SCC25 cells. Cells were treated with 25–100 µg/mL EECRp70 and 25 µM genistein for 24 h. The levels of MMP-2 and VEGF-2 released into the incubation medium were determined using an ELISA kit. Values (mean ± SD) that do not share a common letter are significantly different ($p < 0.05$).

*3.5. Synergy of Antiproliferation between EECRp70 and Cisplatin or Taxol*

The possible synergy in the anti-proliferative effect of SCC25 cells combined with 25 or 50 μg/mL EECRp70 and 100 nM cisplatin or 10 nM taxol is shown in Table 2. The combination of 25 μg/mL EECRp70 with Cis or Ta increased the inhibition of proliferation by 6.52%, 26.90%, and 31.52%, respectively. These values were significantly higher than the expected inhibition values of 20.11% and 19.02% for EECRp70 + Cis and EECRp70 + Ta, respectively. The synergistic effect of the combination treatment was 1.34 fold (EECRp70 + Cis) and 1.72 fold (EECRp70 + Ta) ($p < 0.05$), indicating synergistic effects. To confirm these synergistic effects, SCC25 cells were incubated with a higher concentration of 50 μg/mL EECRp70 with Cis or Ta. The similar synergistic effect resulted in 1.36 fold (EECRp70 + Cis) and 1.54 fold (EECRp70 + Ta) inhibition of proliferation ($p < 0.05$). These results suggest that the synergistic effect of EECRp70 + Ta was higher than EECRp70 + Cis, then we selected EECRp70 in combination 10 nM taxol for further treatment.

**Table 2.** Synergistic effects of antiproliferation of SCC25 cells between EECRp70 and taxol (Ta) or cisplatin (Cis) at 24 h incubation.

| Treatment | Cell Proliferation ($\times 10^4$) [a] | Observed Inhibition (%) | Expected Inhibition (%) | Synergistic Effects [c] | Interaction |
|---|---|---|---|---|---|
| Control | 61.33 ± 1.26 [b] | | | | |
| Taxol 10 nM | 53.67 ± 0.76 | 12.50 ± 1.25 | | | |
| Cis 100 nM | 66.83 ± 1.26 | 13.83 ± 1.64 | | | |
| EECRp70 (25 μg/mL) | 57.33 ± 2.08 | 6.52 ± 3.40 | | | |
| EECRp70 + Cis | 44.83 ± 3.79 | 26.90± 6.17 | 20.11 ± 3.29 | 1.34 | Synergistic |
| EECRp70 + Ta | 42.00 ± 2.18 | 31.52 ± 3.55 | 19.02 ± 4.53 | 1.72 | Synergistic |
| EECRp70 (50 μg/mL) | 49.00 ±0.50 | 20.11 ± 0.82 | | | |
| EECRp70 + Cis | 33.50 ± 1.00 | 45.38 ± 1.63 | 33.70 ± 1.41 | 1.36 | Synergistic |
| EECRp70 + Ta | 25.50 ± 0.50 | 58.42 ± 0.76 | 32.61 ± 1.70 | 1.54 | Synergistic |

[a] SCC25 cells are incubated with 25 or 50 (μg/mL) of EECRp70 with or without 10 nM taxol or 100 nM cisplatin for 24 h. [b] Values are expressed as mean ± SD, n = 3, $p < 0.05$. [c] The synergistic effect is calculated as (control − [EECRp70 + chemotherapy])/((control − EECRp70) + (control − chemotherapy drug)). For value > 1 synergistic, additive 0.5–1.0, antagonistic < 0.5.

*3.6. Effect of EECRp70 on Cell Cycle in SCC25 Cells*

To investigate whether EECRp70 could induce the cell cycle in SCC25 cells, flow cytometric analyses of PI/RNase stained nuclei were performed. As shown in Figure 6, after 24 h of treatment with genistein, EECRp70, and taxol, there was a SubG1 peak, arrest in G2/M, and a decrease in G1 and S phases. Especially, in the group combining EECRp70 and taxol, the subG1% in the cell cycle increased significantly, and the percentage of the population in G1 phase decreased 1.85-fold, S phase decreased 1.54-fold and G2/M phase increased 2.93-fold, compared with EECRp70 alone ($p < 0.05$). Treatment with 25 μM genistein significantly inhibited cell cycle progression in SCC25 cells, resulting in a decrease in G1 and S phases, and an increase in G2/M phase ($p < 0.05$). The subG1% of SCC25 cells in the combination of EECRp70 (25 μg/mL) and taxol (10 nM) did not show a synergistic effect, but an additive effect.

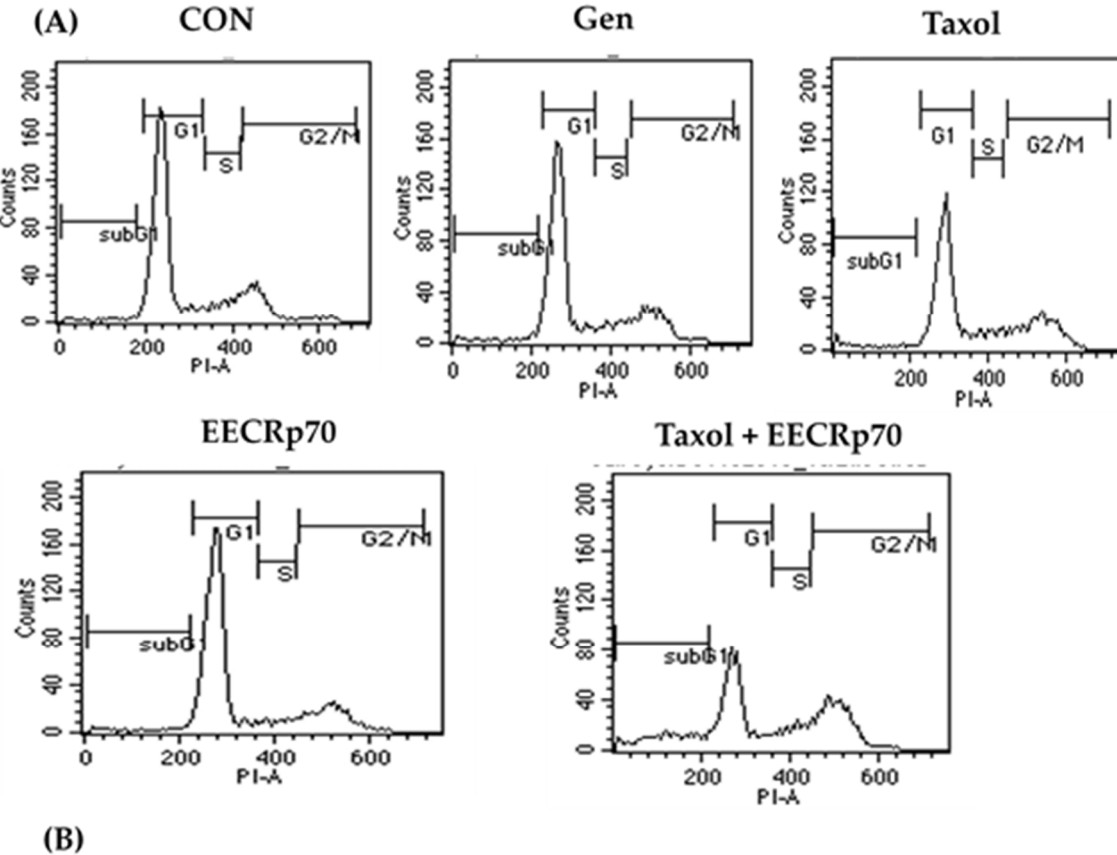

**Figure 6.** Change in cell cycle distribution of SCC25 cells treated with EECRp70. Cells were incubated with 25 μM genistein, 10 nM taxol, 25 μg/mL EECRp70, and 10 nM taxol + 25 μg/mL EECRp70 for 24h. Cells were collected and stained with PI/RNase solution, and the cell cycle was measured by flow cytometry (FACScan). (**A**) Representative cell cycle profiles of EECRp70 treated SCC25 cells. (**B**) Cell cycle of distribution statistics (%). Data as mean ± SD, n = 3, $p < 0.05$ in each column not having the same superscript are significant differences.

### 3.7. Effect of EECRp70 on ROS Level in SCC25 Cells

The DCFH-DA assay was employed to examine the intracellular ROS level in SCC25 cells. As shown in Figure 7, the ROS level increased by 129.45 % in the control. Treatment with 25 μM genistein, 10 nM taxol, and 25 μg/mL EECRp70 reduced the ROS level to 87.78%, 68.29%, and 96.35%, respectively. In addition, the combination of EECRp70 with taxol dramatically decreased the levels of ROS by approximately 46.66% ($p < 0.05$).

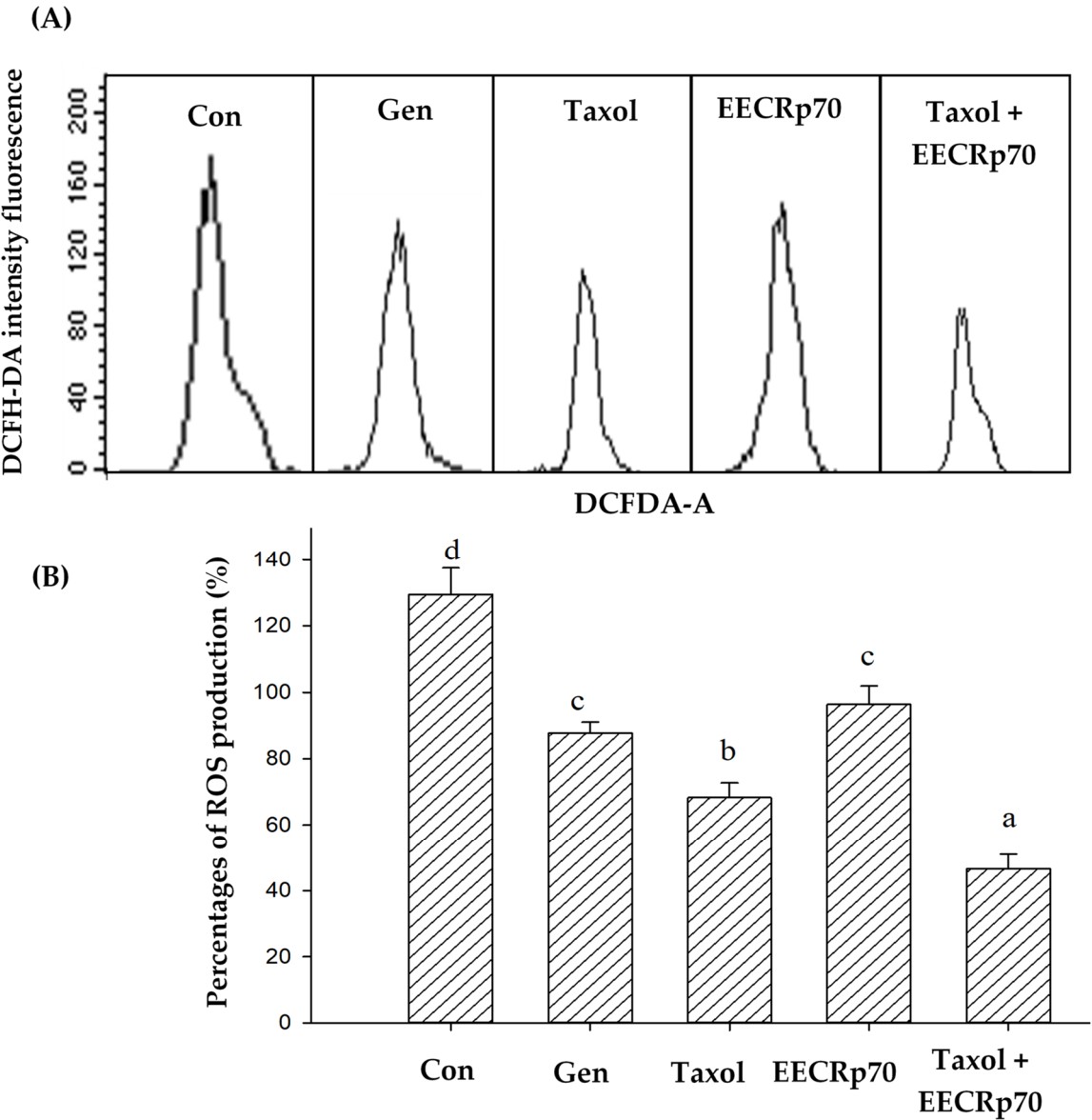

**Figure 7.** Effect of EECRp70 on ROS production. SCC25 cells treated with 25 μM genistein (Gen), 10 nM taxol (Ta), 25 μg/mL EECRp70 and 10 nM taxol + 25 μg/mL EECRp70 incubated for 24 h. Intracellular levels of ROS were measured by DCF the fluorescence intensity using a flow cytometer. (**A**) Cells were collected and incubated with 30 μM DCFH-DA at 37 °C for 1 h in the dark, washed with PBS × 1, and analyzed by flow cytometry. (**B**) Quantitative analysis of the percentage of ROS production in the cells. Values (mean ± SD) that do not have the same letter are significantly different ($p < 0.05$).

### 3.8. Effect of EECRp70 on Migration and Invasion in SCC25 Cells

A Boyden chamber transwell system was used to study the migration and invasion of SCC25 cells 24 h after treatment. Figure 8 shows that the migrated cells were significantly decreased in a dose-dependent manner after SCC25 cells were treated with EECRp70 at concentrations of 25, 50, and 100 μg/mL for 24 h ($p < 0.05$), and the percentage reductions were 9.36%, 19.11%, and 39.88%, respectively. The $IC_{50}$ values of EECRp70 on the migration of SCC25 cells were about 126 μg/mL. At a concentration of 100 μg/mL, the number of migrated SCC25 cells treated with EECRp70 was comparable to that of 25 μM genistein or 10 nM taxol ($p < 0.05$). However, when a combination of 10 nM of taxol and 25 μg/mL

EECRp70 was administered with SCC25 cells, the number of migrated cells was significantly lower than in the group treated alone ($p < 0.05$).

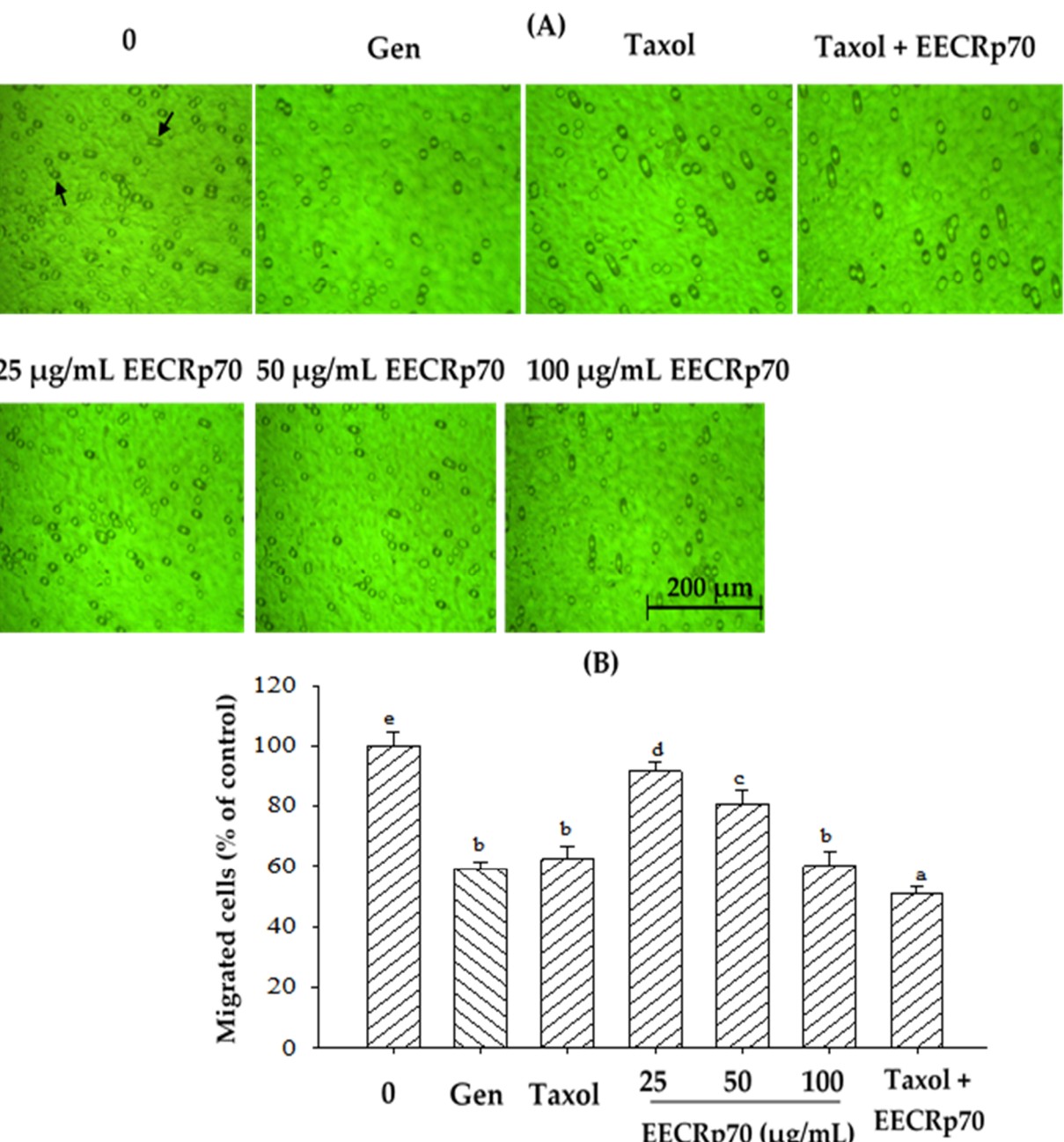

**Figure 8.** Effects of EECRp70 on a transwell migration assay in SCC25 cells. (**A**) Photomicrographs and (**B**) percentages of migrated cells. Cells were incubated with 25 μM genistein, 10 nM taxol, 25–100 μg/mL EECRp70, and 10 nM taxol + 25 μg/mL EECRp70 for 24 h. Photomicrographs were obtained using an Inverted Fluorescence Microscope BestScope-7000B (×200 magnification and scale bar 200 μm). For each replicate, the cancer cells in six randomly selected fields were determined and counted averaged. The Arrowhead indicated the migrated cells. Values (mean ± SD) not labeled with a common letter are significantly different ($p < 0.05$).

As shown in Figure 9, incubation of SCC25 cells with EECRp70 (25, 50, and 100 μg/mL) for 24 h significantly inhibited cell invasion, the inhibition percentages were 9.78%, 18.52%, and 33.26%, respectively, indicating a concentration-dependent effect ($p < 0.05$). The IC$_{50}$ values for EECRp70 on SCC25 cells invasion were approximately 149 μg/mL. The

effect of EECRp70 on invasion inhibition was approximately equivalent to that of 25 μM of genistein (positive control) and 10 nM taxol (a chemotherapeutic agent) ($p < 0.05$). As expected, invasion of SCC25 cells treated with the combination of taxol (10 nM) and ECRp70 (25 μg/mL) significantly decreased compared to cells treated with taxol or EECRp70 alone ($p < 0.05$).

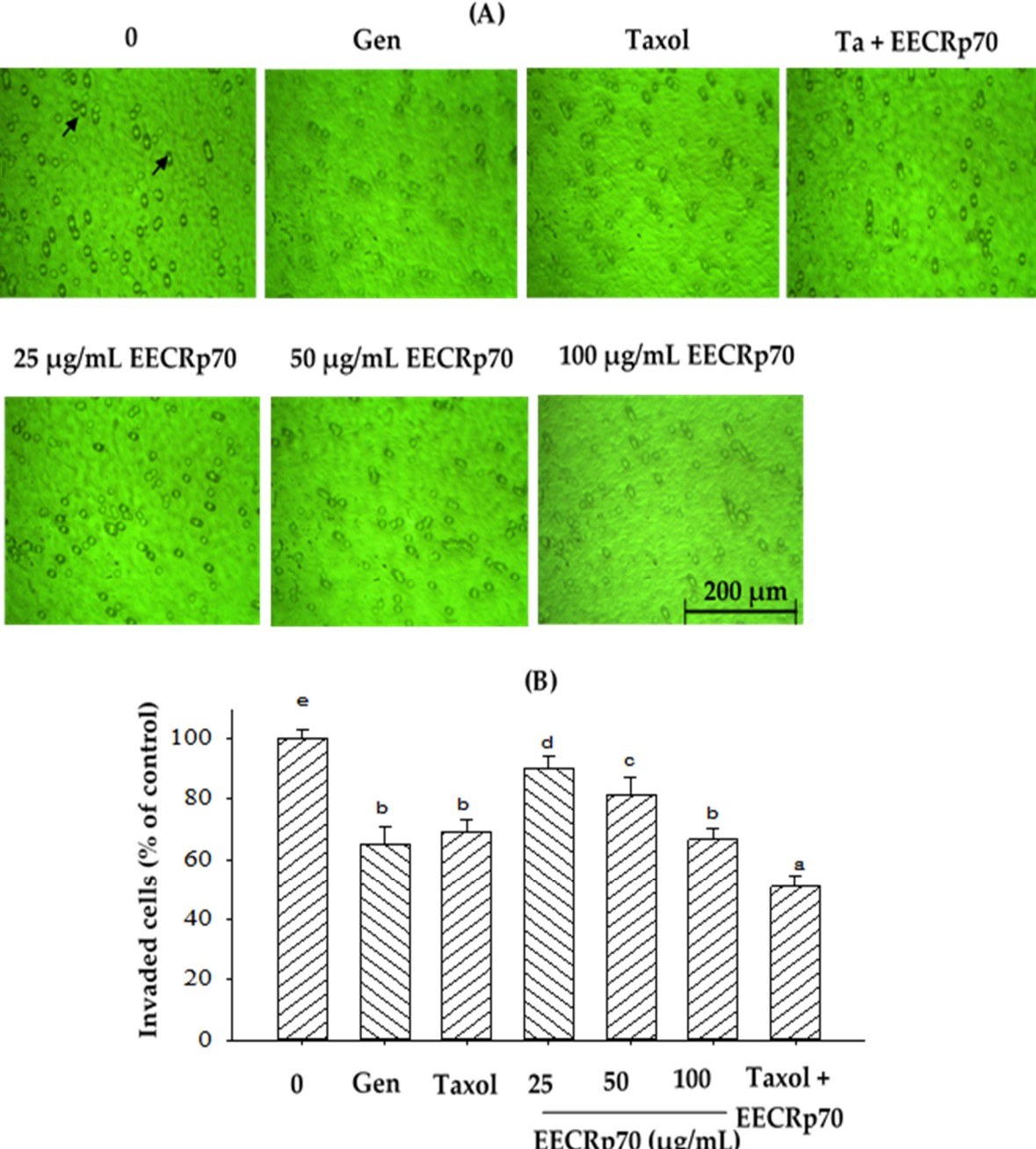

**Figure 9.** Effects of EECRp70 on a transwell invasion assay in SCC25 cells. (**A**) Photomicrographs and (**B**) percentage of invaded cells. Each upper transwell was pre-coated with Matrigel in cold DMEM/F12 before cells were incubated with 25 μM genistein, 10 nM taxol, 25–100 μg/mL EECRp70, and 10 nM taxol + 25 μg/mL EECRp70 for 24 h and calculated the invasion cells of the transwell. Photomicrographs were obtained using an Inverted Fluorescence Microscope BestScope-7000B (×200 magnification and scale bar 200 μm). For each replicate, cancer cells in six randomly selected fields were determined, counted, and averaged. The arrowhead indicates the invaded cells. Values (mean ± SD) not labeled with a common letter are significantly different ($p < 0.05$).

### 3.9. Effect of EECRp70 on Protein Expression in SCC25 Cells

SCC25 cells were treated with 12.5–50 µg/mL EECRp70 for 24 h, protein expressions of iNOS, COX-2, NF-kB (cytoplasm and nuclear) were significantly decreased in a concentration dependent manner ($p < 0.05$) (Figure 10A), the percentage of inhibition of iNOS and COX-2 was about 80% (Figure 10B,C), and NF-kB protein expressions in the cytoplasm and nuclear were 71% and 66%, respectively (Figure 10D,E). In addition, the protein expressions of iNOS, COX-2, and NF-kB of EECRp70 at 25 µg/mL were comparable to those of genistein at 25 µM ($p < 0.05$).

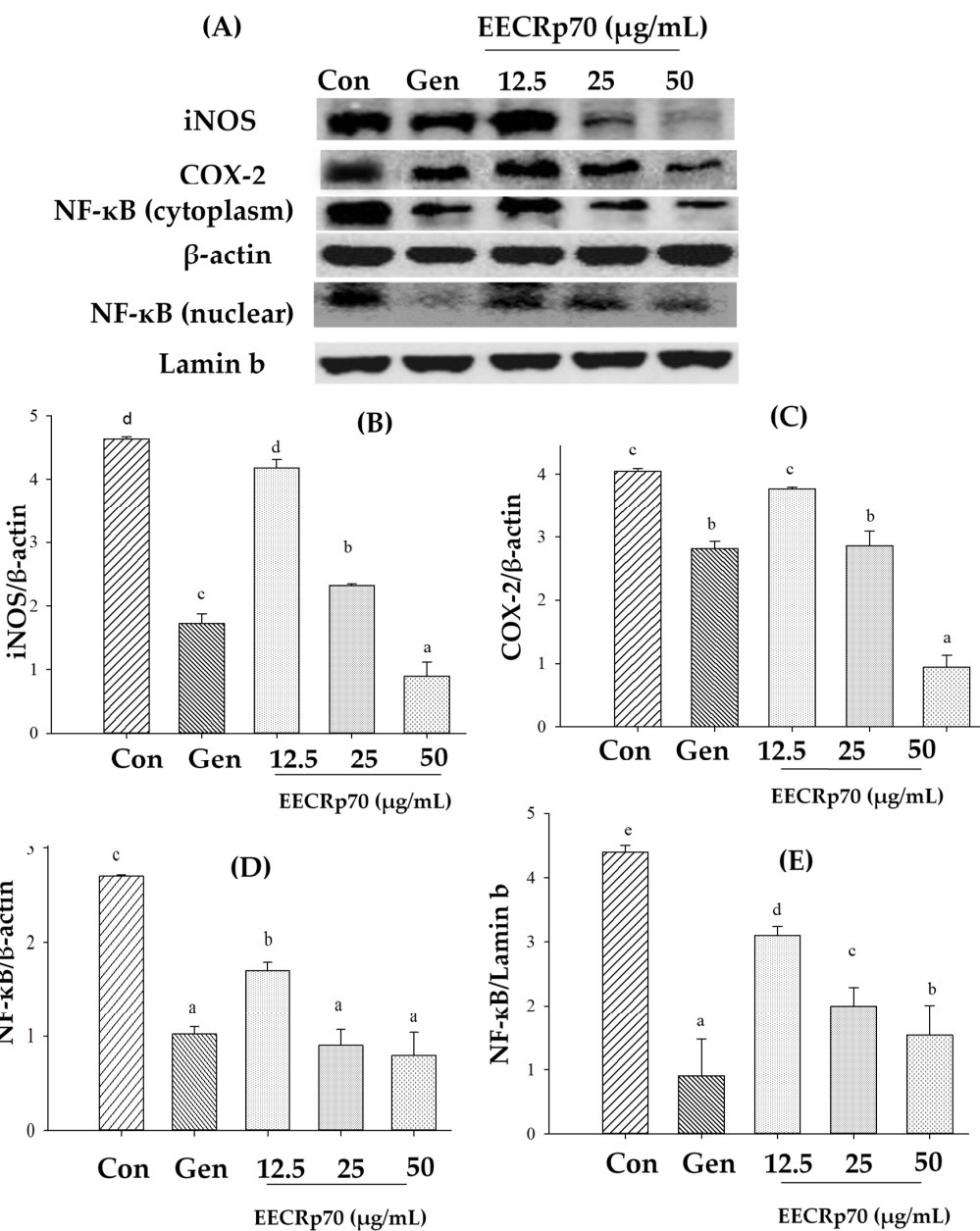

**Figure 10.** Effects of EECRp70 on expression level in SCC25 cells. Cells were treated with EECRp70 (12.5–50 µg/mL) and genistein (25 µM) for 24 h. (**A**) Protein expression of iNOS, COX-2, NF-kB (cytoplasm) and β-actin, or NF-kB (nuclear) and lamin b were detected in cytoplasm and nuclear by Western blotting. Quantitative values of (**B**) iNOS, (**C**) COX-2, (**D**) NF-kB (cytoplasm), and (**E**) NF-kB (nuclear) were analyzed using ImageJ software. Values (mean ± SD) that do not share a common letter are significantly different ($p < 0.05$).

*3.10. Cellular Uptake of Genistein*

As shown in Table 3, the genistein uptake of SCC25 cells treated with 25 µM genistein (6.8 µg/mL) was approximately 1296.67 ng/$10^6$ cells, and the cellular uptake ranged from 5.17% to 10.49% after 3-12 h incubation. The genistein uptake level of SCC25 cells treated with 50 µg/mL of EECRp70 (containing 5.01 µg genistein/mL) was only 251.05 ng/$10^6$ cells, and the cellular uptake was from 0% to 2.71% after 3-12 h incubation. However, genistein uptake of genistein and EECRp70 decreased to 1.65% and 0.83% after 24 h incubation in SCC25 cells. We also analyzed the uptake of cajanol and daidzein in SCC25 cells treated with 50 µg/mL EECRp70. However, neither cajanol nor daidzein was detected in SCC25 cells.

**Table 3.** Determination of cellular uptake of genistein in SCC25 cells.

| Incubation Time (h) | 25 µM Genistein (6.8 µg Genistein/mL) | | 50 µg/mL EECRp70 [1] (5.01 µg Genistein/mL) | |
|---|---|---|---|---|
| | Uptake Level (ng/$10^6$ Cells) | Uptake % | Uptake Level (ng/$10^6$ Cells) | Uptake % |
| 3 | 638.62 ± 18.59 [b2] | 5.17 | ND [3] | ND |
| 6 | 1067.36 ± 11.92 [c] | 8.63 | 167.47 ± 22.96 [a] | 1.81 |
| 12 | 1296.64 ± 67.05 [d] | 10.49 | 251.05 ± 11.12 [c] | 2.71 |
| 24 | 203.50 ± 10.01 [a] | 1.65 | 77.23 ± 12.17 [a] | 0.83 |

[1] EECRp70: sub-fraction in 70% ethanol of 95% ethanol extract from *C. cajan* roots. [2] Values (means ± SD, n = 3, $p < 0.05$) in each column that do not have the same superscript letter are significant differences. [3] ND: not detectable.

## 4. Discussion

*C. cajain* is a widely used traditional plant in many Asian countries including Taiwan. It possesses many bioactive compounds with potential health benefits including anti-inflammatory, antioxidant, anti-microbial, anti-hyperglycemic, antihyperlipidemic, and anti-cancer properties [10,24]. Although the anti-cancer effects of *C. cajain* have been reported in the literature, the major active constituents have not been clearly elucidated. Recently, we showed that EECR95 has the greatest antioxidant and anti-inflammatory effects on macrophages RAW 264.7 cells [14]. In this study, EECR95 was further fractionated to obtain five various sub-fractions as follows: EECRpw, EECRp25, EECRp50, EECRp70, and EECRp95 (under similar conditions of time, particle size, temperature, and liquid/solid ratio). We chose EECRp70 which showed the most potential to induce cytotoxicity with other sub-fractions, which contained the most abundant flavonoids (genistein and cajanol), to evaluate the anti-proliferative and anti-metastatic effects and their possible mechanisms in SCC25 cells. Moreover, both EECRp70 and chemotherapeutic agents (cisplatin or taxol) were co-cultured with SCC25 cells at concentrations that were not cytotoxic (cell viability > 90%) to observe a significant synergistic effect of anti-proliferative and anti-metastatic effects. Therefore, we hypothesize that EECRp70 may be a potent anti-cancer remedy or may increase the sensitivity of anti-cancer drugs to reduce the side effects of chemotherapeutic drugs. Dong et al. (2013) indicated that the combination of a low dose of genistein and daidzein compared with individual soy isoflavone has synergistic preventive effects on human prostate cancer cells by inhibiting cell proliferation and inducing apoptosis via the mechanism that soy isoflavones (daidzein and genistein) can be taken up by cells [36]. Thus, we hypothesize that EECR95 may be an effective anti-cancer agent or may increase the sensitivity of anti-cancer drugs to reduce the side effects of chemotherapeutic drugs.

Taxol and cisplatin are important natural products with known anti-cancer agents, which are used for breast, ovarian, oral, cervical, and lung cancer; however, there are many clinical limitations including dose-limiting side effects and drug resistance [37–43]. The combination of apigenin and paclitaxel (taxol) improved the efficacy of paclitaxel as a chemotherapeutic agent, reduced of the dose of taxol in cancer therapy, mechanically inhibited superoxidase dismutase (SOD), ROS-induced activation of caspase-2, reduced MMP, and induced apoptosis in cancer cells [40]. Sui et al. demonstrated that cajanol

induces paclitaxel efflux in A2780/taxol ovarian cancer cell lines by down-regulating permeability glycoprotein (P-gp) expression, and cajanol inhibits P-gp transcription and translation via the PI3K/Akt/NF-κB pathway [44]. Therefore, we hypothesize that our sub-fraction EECRp70 and anti-cancer drugs (taxol and cis) may synergistically inhibit the proliferation of SCC25 cells. Our results showed that EECRp70 inhibited the proliferation of oral cancer cells SCC25 in a dose and time-dependent manner. We then investigated the synergistic abilities of EECRp70 and taxol or cisplatin on SCC25 cells.

In this study, we demonstrated that the combination of EECRp70 at non- or sub-cytotoxic concentrations with taxol or cisplatin exhibited synergistic anti-proliferative effects. Moreover, the combination of EECRp70 with taxol results in a significant increase in migrated and invaded cells in SCC25 compared to cells treated alone ($p < 0.05$). Due to the various side effects and resistance to chemotherapy agents, pharmacists at that time used non-toxic natural compounds as adjuvants to chemotherapeutic agents to reduce the adverse effects. Thus, we suggested that EECRp70 has potential as an adjuvant for chemotherapy.

Cell growth and proliferation of mammalian cells are mediated by cell cycle processing [13]. Loss of cell cycle checkpoint control at SubG1, G1, S, and G2/M phases underlies the aberrant proliferation of cancer cells. Several studies have shown that the flavonoid, apigenin, cajanol (from CR), and quercetin arrest G1 and G2/M phases in SCC25 cells, and genistein arrest S and G2/M phases in PC-3 and LNCaP cells [15,19,43,45]. The results show that dose-dependently, EECRp70 alone and combined with taxol caused cell cycle increases in SubG1 and arrest in G2/M phases as well as a decrease in G1 and S phases of SCC25 cells. The distribution of cells in different cell cycle phases proves the multi-effect of EECRp70 on human cancer cells. Thus, the cell apoptosis and inhibition of cell proliferation induced by EECRp70 were associated with the induction of cell cycle arrest. The results of this suggest that this combination can inhibit cell growth and proliferation, reduce the side effects of chemotherapeutic drugs, and increase the survival rate of patients.

Jhou et al. pointed out that ROS plays an important role in the progression of many different diseases, such as carcinogenesis, metastasis, and angiogenesis [46]. ROS is also involved in signal transduction of tumor cell proliferation, inflammation, and immune responses. In addition, the release of ROS has been shown to activate NF-kB signaling and NF-kB transcription factors to regulate the expression of key genes for cell proliferation and survival. For example, NF-kB activation may be required for other cell lines such as human cancer, cervical, and ovarian cancer [4,44,47]. In the study, it was shown that the reduction of ROS production by EECRp70. We also found that EECRp70 inhibited the expression of NF-kB protein in the cytoplasm and nuclear and decreased the expression of iNOS and COX-2. EECRp70 decreased the intercellular ROS, which may inhibit the metastasis of SCC25 by reducing the binding activity of the NF-kB pathway, so EECRp70 may also play a role in targeting proliferation and the cell cycle.

Immunohistochemical studies have indicated that the MMP family is involved in basement membrane proteolysis and tissue invasion in OSCC [35]. Expression of MMP-2 has been observed in invasive and metastatic cases of OSCC [48–50]. Moreover, increased MMP-2 secretion and expression is associated with decreased extracellular matrix staining in OSCC, suggesting that MMP-2 promotes matrix breakdown (3,8). The MMPs lead to tumor metastasis including migration, invasion, and angiogenesis [47]. Inhibition of MMP expression or enzyme activity can be used as an early target to prevent cancer metastasis MMP-2 and VEGF-2 are involved in the invasive metastatic potential of tumor cells [51]. We found that pretreatment inhibited the migration and invasion of SCC25 cells by inhibiting the activity of MMP-2 and VEGF-2. Thus, these results suggest that EECRp70 can inhibit MMP-2 and VEGF-2 signaling pathways to exert its anti-metastatic effects such as inhibiting proliferation, migration, and invasion of SCC25 cells in vitro.

In this study, based on fractionation and RP-HPLC analysis, three phenolic compounds (daidzein, cajanol, and genistein) were detected; EECRp70 is the strongest fraction to explore cytotoxicity against SCC25 cells. Genistein (4′,5,7-trihydroxyisoflavone, $C_{15}H_{10}O_5$),

is a flavonoid with phytoestrogen activity, mainly derived from Fabacea such as *Lupinus albus* L. (lupine), *Vicia faba* L (fava bean), *Glycine max* (L.) Merr. (soybeans), *Pueraria lobata* (Willd.) Ohwi (kudzu), *Psoralea corylifolia* L. (Psoralea), and *Cajanus cajan* (L.) Millsp (pigeon pea [52,53]. Genistein possesses antioxidant, anti-inflammatory, and anti-cancer properties [53–56]. Genistein also has many health-promoting benefits, such as reducing the risk of cardiovascular disease, alleviating symptoms of menopause, anti-cancer properties, obesity, diabetes, and anxiety, and protecting against osteoporosis [57,58]. In addition, genistein also exerts apparent anti-tumor properties by inhibiting the proliferation of MCF-7 and HT-29 cells through several molecular mechanisms that include the induction of apoptosis, transcription, proteins, and enzymes [59]. Another flavonoid, cajanol (5-hydroxy-3-(4-hydroxy-2-methoxyphenyl)-7-methoxychroman-4-one, $C_{17}H_{16}O_6$), is a flavonoid derived from the root of *C. cajan*, which exhibits several pharmacological activities such as anti-bacterial, anti-fungal, anti-malarial, and anti-tumor activities [60]. Cajanol exerts anti-cancer effects by inhibiting cell growth, arresting the cell cycle, and inducing apoptosis in human MCF-7 cells [15]. Sui et al. demonstrated that cajanol inhibits paclitaxel efflux in A2780/taxol ovarian cancer cell lines by down-regulating permeability glycoprotein (P-gp) expression, and cajanol inhibits P-gp transcription and translation via the PI3K/Akt/NF-κB pathway [44].

Spencer et al. reported that flavonoids as chemopreventive agents must fulfill the following three mechanisms: (1) inhibition of metabolic activation of carcinogens, (2) inhibition of tumor cell proliferation by inactive or down-regulative pro-oxidant enzymes or signal transduction enzymes, and (3) induced death of tumor cells (apoptosis). Spencer et al. pointed out that the level of uptake and the range of metabolism for flavonoids is an important step for these mechanisms in cancer cells [61]. In addition, Salucci et al. reported that polyphenols (such as gallic acid, epicatechin, epigallocatechin gallate, and quercetin) were significantly taken up by human colon adenocarcinoma Caco-2 cells during 24 h incubation [62]. Our results also showed that when pure genistein and EECRp70 were added to SCC25 cells and cultured for 12 h, the uptake content of genistein reached the maximum value (10.49 and 2.71%), indicating that the genistein contained in EECRp70 would indeed enter SCC25 cells for molecular regulation, so genistein should be one of the main indicator components in EECRp70. However, the uptake percentage of pure genistein is higher than that of genistein in EECR70, which is likely to be related to the coexisting competitive uptake effect caused by the presence of other polyphenols (such as biochanin A) in EECRp70 [14].

Mechanistically, the strongest fraction-EECRp70 fractionated from EECR95 exerted potent anti-proliferation and anti-metastasis effects is due to (1) cell cycle arrest (arrest in G2/M phases) and (2) inhibition of MMP-2 and VEGF-2 secretion via attenuate ROS formation by blocking of NF-kB/iNOS/COX-2 signaling pathway in SCC25 cells. Moreover, EECRp70 inhibited cell proliferation, metastasis, and cell cycle arrest in SCC25 cells in synergy with chemotherapy agents. Thus, EECRp70 can be used as an adjuvant in cancer treatment to reduce the side effects of chemotherapeutic agents. Furthermore, cellular uptake of genistein in EECRp70 elucidate that genistein is the main active compound in EECRp70 that affects the proliferative and metastatic effects of OSCC (Figure 11).

To our knowledge, there is no published article on the safe dose of EECRp70, but we recently reported in a rat study that the NOAEL for EECR is approximately 1 g/kg of body weight [63] which is equivalent to approximately 972 mg/60 kg person/day by extrapolation from rats to humans [24].

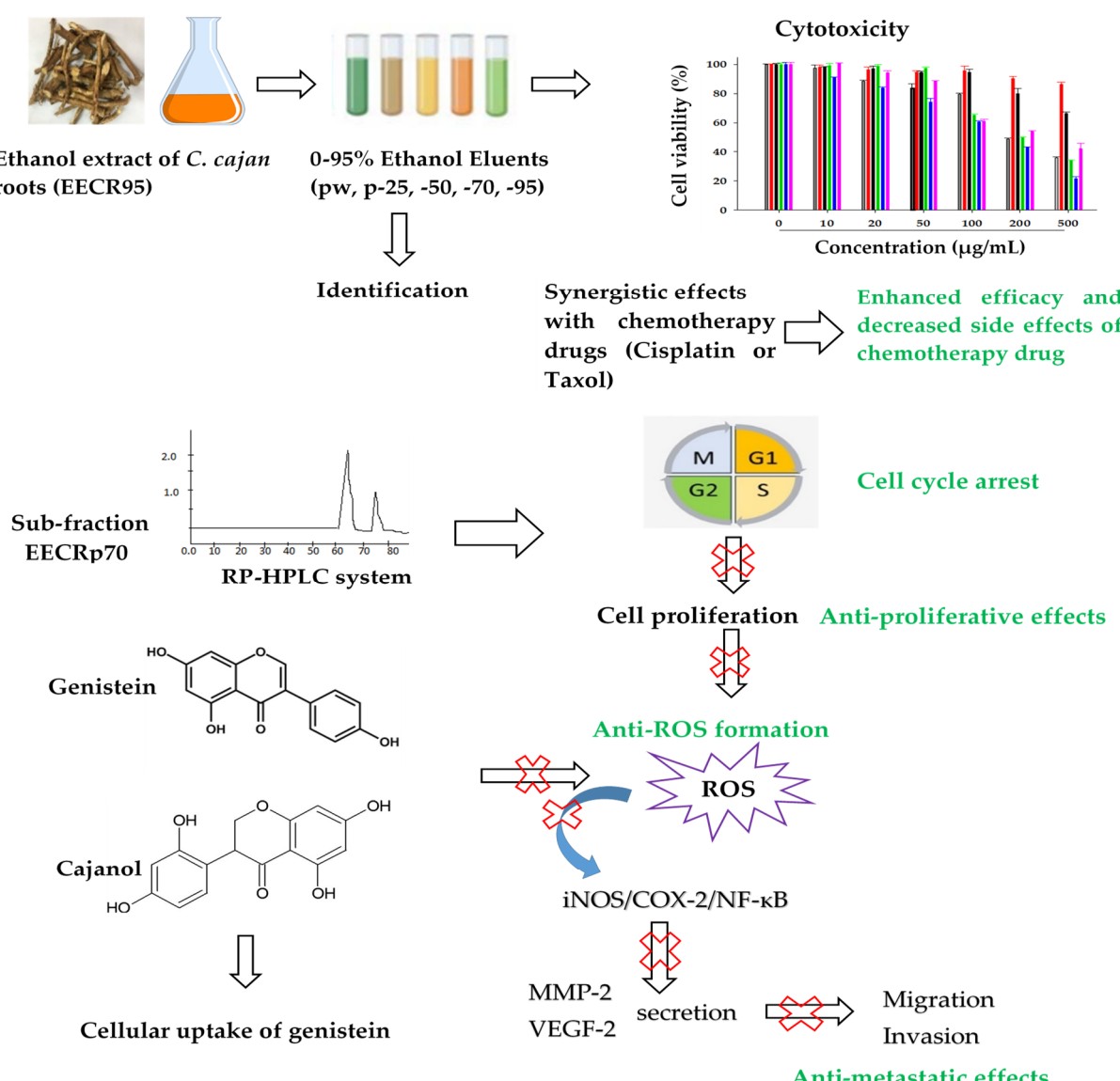

**Figure 11.** Schematic summaries of the anti-proliferative and anti-metastasis effects of EECRp70 on SCC25 cells. Crude 95% ethanol extract from *C. Cajan* roots (EECR95) was fractionated (pw, p25–95). The strongest fraction was selected for cytotoxicity and identification of bioactive compounds using RP-HPLC systems. Two kinds of flavonoids (genistein and cajanol) were found high in sub-fraction EECRp70, which showed beneficial effects. In addition, EECRp70 synergized with chemotherapeutic agents, reduced cell proliferation, inhibited cell metastatic effects, caused cell apoptosis, and finally cancer cell death; cellular uptake of genistein could cause the EECRp70 proliferative and metastatic effects of oral squamous cell carcinoma cells.

## 5. Conclusions

In conclusion, the present study demonstrated for the first time the anti-oral cancer potential of EECRp70, the sub-fraction of EECR95, which possesses potent anti-proliferative and anti-metastatic activity in human oral cancer SCC25 cells. EECRp70 contains potent phytochemicals (such as, cajanol, daidzein, and genistein) that could be used as daily dietary supplements or as adjuvant therapy in the treatment of oral cancer. More in-depth studies are needed to determine the safety threshold of EECRp70 as a potential adjuvant to chemotherapy.

**Author Contributions:** Conceptualization, L.-G.H.; methodology and software, P.-H.L.; formal analysis, T.-L.T.V. and S.-E.Y.; investigation, T.-L.T.V.; data curation, T.-L.T.V. and C.-L.C.; writing—original draft preparation, T.-L.T.V.; writing—review and editing, T.-Y.S.; supervision, T.-Y.S.; project administration, T.-Y.S. All authors have read and agreed to the published version of the manuscript.

**Funding:** This research was funded by the National Science and Technology Council of the Republic of China (ROC), Taiwan, for financially supporting this research under contract No. MOST 106-2320-B-212-002, and partly funded by the Taichung Veterans General Hospital Project, Da-Yeh University Taiwan, R.O.C. under contract No. TCVGH-DYU1088304 and Ministry of Education awards and subsidies to Chienkuo Technology University (CTU-110-RP-BS-001-010-A).

**Institutional Review Board Statement:** Not applicable.

**Data Availability Statement:** Not applicable.

**Conflicts of Interest:** The authors declare no conflict of interest.

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
