# Peer review of "Anti-Proliferative and Anti-Metastatic Effects of Ethanol Extract from Cajanus cajan (L.) Millsp. Roots and its Sub-Fractions in Oral Squamous Cell Carcinoma"

_agriculture, doi:10.3390/agriculture12121995_

Round 1

Reviewer 1 Report

Manuscript ID: agriculture-1988677
Type of manuscript: Article
Title: Anti-proliferative and anti-metastatic effects of ethanol extracts of Cajanus cajan (L) Millsp. roots and its sub-fractions in Oral squamous cell carcinoma
Authors: Thuy‑Lan Thi Vo, Shu‑Er Yang, Liang-Gie Huang, Po-Hsien Li, Chien‑Lin Chen, Tuzz‑Ying Song

Dear Editor

Thank you for inviting me to review the manuscript

“ Anti-proliferative and anti-metastatic effects of ethanol extracts of Cajanus cajan (L) Millsp. roots and its sub-fractions in Oral squamous cell carcinoma”.

Below, I am giving are some comments that will improve the quality of this manuscript.

Best regards

Recenzent

Title and abstract

1. Lines 2 and 15 – There should be a dot after L: Cajanus cajan (L.)

2. Line 3 – Oral squamous cell carcinoma should be written with small “o”: oral squamous cell carcinoma

3. Line 15 – An abbreviation CR should be described precisely in place, where they are used at the first time.

Abstract

4. The abstract should include the following elements: (a) place the introduction addressed in a broad context, (b) highlight the purpose of the study, (c) describe briefly the main methods applied, (d) summarize the main findings, (e) and indicate the main conclusions.

Keywords

5. Keywords should not contain words appearing in the title of the study.

6. Lines 30-31 – It is better when keywords are nouns than adjectives: synergistic – synergism

Introduction

7. Line 34 – The word and is not needed … cancers), and causing …

8. Lines 38-39 – The words Traditional treatments … in OSCC are not a sentence as there is no verb and it is hard to understand their meaning without appropriate correction. I also propose to add citation of at least one newer article about this neoplasm (e.g. this below which is five years newer than these from 2017):

Oral squamous cell carcinoma – clinical characteristics, treatment, and outcomes in a single institution retrospective cohort study. Janiak-Kiszka J, Nowaczewska M, Kaźmierczak W. Otolaryngol Pol. 2022 Feb 22;76(3):12-17. doi: 10.5604/01.3001.0015.7567.

9. Lines 43-44 – The words Plants namely … semi-arid tropics are not a sentence as there is no verb and it is hard to understand their meaning without correction.

10. Lines 62-64 – The words Many researchers … human cancer cell line are not a sentence as there is no verb and it is hard to understand their meaning without correction.

11. Line 67 – The word study have to be in its past form studied.

12. Line 71 – I propose to write anti-proliferative effect instead of anti-proliferation.

13. Please formulate a specific aim of the study.

Materials and methods

14. Lines 87-88 – The words Extracellular matrix … (Sigma-Aldrich Corp) are not a sentence as there is no verb and it is hard to understand their meaning without correction.

15. Section 2.6 – Grammar mistakes should be corrected (especially past form of verb and sentence construction).

Results

16. Please eliminate double or triple citation the same literature item in adjacent phrases or tasks (lines 58 - 61, 367 – 368, 391 – 396. 457 – 462 itp.).

Table 2

17. Please eliminate in the table minus sign "-",

18. Please insert relevant information adopted for this type of research

19. In figures 7 and 8, in the photomicrographs, please insert magnifications (Bars).

20. Line 244 – I propose to write anti-proliferative effect instead of anti-proliferation.

21. Line 275 – It should be synergistic effect, not synergist effect. The word synergist is a noun, not adjective.

22. Line 301 – The word for should be written with small letter “f”.

23. Please formulate the aim of the study.

Discussion

24. As per the guidelines for authors please make a correction l. 367 -368 … Yang et al. (2020) ….. Yang et al. (2022) [24] …”

25. Lines 364-366 – This sentence is not precise as activity is compared with contents. Moreover the word most (in a phrase most potent) is not needed or the construction of this sentence should be changed.

26. Line 372 – There is no year of Dong’s publication.

27. Lines 380-381 – I propose to write various instead of different.

28. Line 383 – I propose to write the biggest potential instead of the most potent as it should be a phrase with a noun described by a adjective in its highest degree.

29. Line 383 – I propose to write in comparison with instead of than.

30. Lines 460-462 – This sentence is not precise because of grammatical mistake.

31. Line 469 – It is better to write percentage than %.

Conclusions

32. Figure 10 should be in Results part of the article, not in Conclusions.

References

33. Check the list of references and the entire text so that they follow the guidelines for authors.

34. Line 500 „1941-1949”, line 508 „856–966) e.t.c.

35. Line 496 „Sci Rep”, line 521 „J. Complement. Integr. Med.” e.t.c.

36. Line 525 „2009, 117(1), 152-159” itd.

37. Line 540 „Cajanus cajan” e.t.c.

38. Lines 497,498; 511-514; 542, 543; 554,555; 567, 568; 569,570; 588, 589; 595, 596; 600, 601 „Insulin Sensitizer and Antihyper-li-pidemic Effects of Cajanus cajan (L.) Millsp. Root in Methylglyoxal‑Induced Diabetic Rats. Chin J Physiol 2022, 65, 125-135 „ e.t.c.

39. Line 568 „Academic Journal of Cancer Research” e.t.c.

Author Response

Reviewer 1

Title and abstract

  1. Lines 2 and 15 – There should be a dot after L: Cajanus cajan (L.)

Ans: Thank you for pointing out our mistake. We corrected it page 1, line 3, 15 highlight in red.

  1. Line 3 – Oral squamous cell carcinoma should be written with small “o”: oral squamous cell carcinoma

Ans: We corrected it page 1, line 3 highlight in red

  1. Line 15 – An abbreviation CR should be described precisely in place, where they are used at the first time.

Ans: CR is the abbreviation of C. cajan (L.) Millsp roots, we have changed in full name page 1, line 15 highlight in red.

Abstract

  1. The abstract should include the following elements: (a) place the introduction addressed in a broad context, (b) highlight the purpose of the study, (c) describe briefly the main methods applied, (d) summarize the main findings, (e) and indicate the main conclusions.

Ans: We have rewritten in abstract page 1, line 27-30 highlight in red 

Keywords

  1. Keywords should not contain words appearing in the title of the study.

Ans: We changed words in keywords page 1, line 31-32 highlight in red.

  1. Lines 30-31 – It is better when keywords are nouns than adjectives: synergistic – synergism

Ans: Thank you for your reminding. We have changed adjectives keywords to nouns page 1, line 31 highlight in red.

Introduction

  1. Line 34 – The word and is not needed … cancers), and causing …

Ans: We have deleted “cancer” and “causing” words.

  1. Lines 38-39 – The words Traditional treatments … in OSCC are not a sentence as there is no verb and it is hard to understand their meaning without appropriate correction.

I also propose to add citation of at least one newer article about this neoplasm (ex. this below which is five years newer than these from 2017):

Janiak-Kiszka J, Nowaczewska M, Kaźmierczak W. Otolaryngol Pol. Oral squamous cell carcinoma – clinical characteristics, treatment, and outcomes in a single institution retrospective cohort study. 2022 Feb 22;76(3):12-17. doi: 10.5604/01.3001.0015.7567.

Ans: - We increase the verb in the sentence page 1, line 38-40 highlight in red.

- We also added the reference [3] the reviewer suggested page 1, line 36 highlight in red.

- In reference [3] Kazmierczak, W.; Janiak-Kiszka, J.; Nowaczewska, M. Oral squamous cell carcinoma – clinical characteristics, treatment, and outcomes in a single institution retrospective cohort study. Otolaryngol Pol 2022, 76, 12-17, page 20, line 537-538 highlight in red.

  1. Lines 43-44 – The words Plants namely … semi-arid tropics are not a sentence as there is no verb and it is hard to understand their meaning without correction.

Ans: We have changed this sentence in page 1, line 44-45 highlight in red.

  1. Lines 62-64 – The words Many researchers … human cancer cell line are not a sentence as there is no verb and it is hard to understand their meaning without correction.

Ans: We have corrected the sentence in page 2, line 52-54 highlight in red.

  1. Line 67 – The word study have to be in its past form studied.

Ans: We have corrected the sentence in page 2, line 76 highlight in red.

  1. Line 71 – I propose to write anti-proliferative effect instead of anti-proliferation.

Ans: We have changed “anti-proliferation” to “anti-proliferative effect” in page 2, line 74.

  1. Please formulate a specific aim of the study.

Ans: We added the aim of study in page 2, line 64-69 highlight in red.

Materials and methods

  1. Lines 87-88 – The words Extracellular matrix … (Sigma-Aldrich Corp) are not a sentence as there is no verb and it is hard to understand their meaning without correction.

Ans: We have added verb in the sentence, page 2, line 89 highlight in red.

  1. Section 2.6 – Grammar mistakes should be corrected (especially past form of verb and sentence construction).

Ans: We have corrected grammar mistakes in page 4, line 133-140 highlight in red.

Results

  1. Please eliminate double or triple citation the same literature item in adjacent phrases or tasks (lines 58 - 61, 367 – 368, 391 – 396. 457 – 462 itp.).

Ans: We were eliminated double or triple citation the same literature item in adjacent phrases or tasks in lines 45-64, 376– 385, 447-454, 480-484

Table 2

  1. Please eliminate in the table minus sign "-",

Ans: We have deleted “-“in table 2, page 10

  1. Please insert relevant information adopted for this type of research

Ans: We have put in the references [Song et al. and Meyer et al.] of the method for the calculation of the synergistic or antagonistic effects in section 2.4 page 4, line 119-124 highlight in red.

  1. In figures 7 and 8, in the photomicrographs, please insert magnifications (Bars).

Ans: We added scale bar in the photomicrographs in figures 8 and 9, page 13, 14.

  1. Line 244 – I propose to write anti-proliferative effect instead of anti-proliferation.

Ans: We have changed to “anti-proliferation“ in anti-proliferative page 10, line 250 highlight in red.

  1. Line 275 – It should be synergistic effect, not synergist effect. The word synergist is a noun, not adjective.

Ans: We changed “synergistic effect” page 10, line 281 highlight in red.

  1. Line 301 – The word for should be written with small letter “f”.

Ans: Thank you for your suggestion. This letter is the first word at the beginning of the sentence, so the "f" remains capitalized page 13, line 309-310 highlight in red.

  1. Please formulate the aim of the study.

Ans: We have been corrected. The aim is added as follows: the aim of this study was to explore the anti-proliferative and anti-metastatic effects of 95% ethanol extracts of C. cajan roots (EECR95) and its sub-fractions in OSCC, and its possible mechanisms. In addition, we also explored the synergistic effect of combination of EECRp70 and chemotherapeutic drugs (cisplatin or taxol) in inhibiting OSCC cell proliferation to realize whether EECR can be used as an adjuvant for cancer treatment to reduce the side effects of chemotherapeutic drugs, page 2, line 65-70 highlight in red.

Discussion

  1. As per the guidelines for authors please make a correction l. 367 -368 … Yang et al. (2020) ….. Yang et al. (2022) [24] …”

Ans: We made corrections the citations in accordance with the journal's author guidelines.

  1. Lines 364-366 – This sentence is not precise as activity is compared with contents. Moreover the word most (in a phrase most potent) is not needed or the construction of this sentence should be changed.

Ans: Thank you for your suggesting. We have re-written this sentence page 16, line 372-375 highlight in red.

  1. Line 372 – There is no year of Dong’s publication.

Ans: We added year “2013” of Dong’s publication page 17, line 392 highlight in red

  1. Lines 380-381 – I propose to write various instead of different.

Ans: We've changed to "various" in page 16, line 382 highlight in red.

  1. Line 383 – I propose to write the biggest potential instead of the most potent as it should be a phrase with a noun described by an adjective in its highest degree.

Ans: We changed to “the biggest potential” in page 17, line 384 highlight in red.

  1. Line 383 – I propose to write in comparison with instead of than.

Ans: Thank you for your suggestion. We corrected it page 17, line 385.

  1. Lines 460-462 – This sentence is not precise because of grammatical mistake.

Ans: We have changed this sentence page 18, line 479-481 highlight in red.

  1. Line 469 – It is better to write percentage than %.

Ans: We corrected it page 18, line 488 highlight in red.

Conclusions

  1. Figure 10 should be in Results part of the article, not in Conclusions.

Ans: Thanks to the reviewer, we have changed Figure 11 to the discussion section, page 19 line 503.

References

  1. Check the list of references and the entire text so that they follow the guidelines for authors.
  2. Line 500 „1941-1949”, line 508 „856–966) e.t.c.
  3. Line 496 „Sci Rep”, line 521 „J. Complement. Integr. Med.” e.t.c.
  4. Line 525 „2009, 117(1), 152-159” itd.
  5. Line 540 „Cajanus cajan” e.t.c.
  6. Lines 497,498; 511-514; 542, 543; 554,555; 567, 568; 569,570; 588, 589; 595, 596; 600, 601 „Insulin Sensitizer and Antihyper-li-pidemic Effects of Cajanus cajan (L.) Millsp. Root in Methylglyoxal‑Induced Diabetic Rats. Chin J Physiol 2022, 65, 125-135 „ e.t.c.
  7. Line 568 „Academic Journal of Cancer Research” e.t.c.

Ans: We have checked the reference in MS carefully and corrected all mistakes in the list of references follow the guidelines for authors.

Reviewer 2 Report

I have carefully read the manuscript entitled “Anti-proliferative and anti-metastatic effects of ethanol extracts of Cajanus cajan (L) Millsp. roots and its sub-fractions in Oral squamous cell carcinoma” by Vo et al. I found this topic interesting but I have few concerns related to the research article. I am asking authors to revise the manuscript carefully considering my comments for possible publication in “Agriculture”. I have given my comments so that authors to rethink and improve the quality of the manuscript:

Abstract:

The main feature of the research study and how it can impact future research should be highlighted in the abstract.

Introduction:

Authors must give some of the information regarding C. cajan species initially in introduction and how this species is economical important and give some information about production aspects.

I have found the manuscript with self-duplications. Some of the part mentioned in introduction is overlapping with that of discussion part. I suggest to restructure the manuscript so that self -duplication can be avoided.

I also observed that ratio of recent references is very less. Authors must include recent references in introduction and discussion part of the manuscript.

Material and methods:

Authors must add a flow diagram showing methodology followed in the experimentation and also show the analysis performed at each stage. This flow diagram will definitely improve the readability of the manuscript.

Result and Discussion:

Authors have presented their finding in well manner with appropriate discussions but I still not satisfied with recent works (2020-2022) carried out on similar area. Most of the cited articles in the discussion section are not sufficient. It is suggested to improve the manuscript considering this specific aspect. I wish to see a revised version of the manuscript with a good discussion throughout the result and discussion section with recent. Similarly for other sections the reason of particular findings is not discussed in sufficient way.

Following article will definitely help to improve discussion part: https://doi.org/10.3390/antiox10091358; https://doi.org/10.3389/fmolb.2021.649395

Way forward to future research must be mentioned in the conclusion.

Moreover, English language needs minor improvements. Punctuations and spacing also need to be checked carefully throughout the manuscript.

Figure 10: This figure should be supplemented with self-explanatory legend. Currently it is not reader friendly.

If authors can take serious efforts to improve the manuscript, I will be happy to re-review the manuscript.

Author Response

Response to reviewer comments and suggestions

Reviewer 2

I have carefully read the manuscript entitled “Anti-proliferative and anti-metastatic effects of ethanol extracts of Cajanus cajan (L) Millsp. roots and its sub-fractions in Oral squamous cell carcinoma” by Vo et al. I found this topic interesting but I have few concerns related to the research article. I am asking authors to revise the manuscript carefully considering my comments for possible publication in “Agriculture”. I have given my comments so that authors to rethink and improve the quality of the manuscript:

Abstract:

  1. The main feature of the research study and how it can impact future research should be highlighted in the abstract.

Ans: We have rewritten in abstract, page 1, line 27-30 highlight in red.

Introduction:

  1. Authors must give some of the information regarding C. cajanspecies initially in introduction and how this species is economical important and give some information about production aspects.

Ans: We have added some of the information regarding C. cajan in introduction page 1-2, line 45-64 highlight in red.

  1. I have found the manuscript with self-duplications. Some of the part mentioned in introduction is overlapping with that of discussion part. I suggest to restructure the manuscript so that self -duplication can be avoided.

Ans: Thanks for reviewer’s reminding. We have re-structured the manuscript to avoid self -duplication.

  1. I also observed that ratio of recent references is very less. Authors must include recent references in introduction and discussion part of the manuscript.

Ans: We have added recent references in introduction and discussion part of the manuscript [3] page 1 line 36; [11] page 1 line 47; [52] page 18 line 461; [58] page 18 line 464 highlight in red.

Material and methods:

  1. Authors must add a flow diagram showing methodology followed in the experimentation and also show the analysis performed at each stage. This flow diagram will definitely improve the readability of the manuscript.

Ans: Thanks for your valuable comments. We have added a flow diagram showing methodology followed in the experimentation and show the analysis performed at each stage in section 2.2, page 3.

Result and Discussion:

Authors have presented their finding in well manner with appropriate discussions but I still not satisfied with recent works (2020-2022) carried out on similar area. Most of the cited articles in the discussion section are not sufficient. It is suggested to improve the manuscript considering this specific aspect. I wish to see a revised version of the manuscript with a good discussion throughout the result and discussion section with recent. Similarly for other sections the reason of particular findings is not discussed in sufficient way.

  1. Following article will definitely help to improve discussion part: https://doi.org/10.3390/antiox10091358; https://doi.org/10.3389/fmolb.2021.649395

Ans: Thank you for your suggestion. We have revised the Discussion to refer to the paper format suggested by the reviewers.

  1. Way forward to future research must be mentioned in the conclusion.

Ans: We added way forward to future research in the conclusion page 20, line 515-518 highlight in red.

  1. Moreover, English language needs minor improvements.Punctuations and spacing also need to be checked carefully throughout the manuscript.

Ans: Thank you for the reviewer's suggestion, we have carefully checked the entire manuscript for English grammar, punctuation and spacing.

  1. Figure 10: This figure should be supplemented with self-explanatory legend. Currently it is not reader friendly.

Ans: We have made a detailed supplement to the legend of Figure 11 for the convenience of readers (the legend of Figure 11 in page 19, line 504-511 highlight in red).

  1. If authors can take serious efforts to improve the manuscript, I will be happy to re-review the manuscript.

Ans: Thanks to reviewer for the valuable suggestions, we will do our best to revise the manuscript.

Reviewer 3 Report

The overall manuscript is written very well. However, before acceptance author should improve these mistakes.

line 15, rewrite, "commonly known as pigeon pea"

Line 15:  what CR?

LINE 28-29, Rewrite concluding lines, clealry brief in two or three lines.

line 43, same line write in abstract ,, remove and start sentence from pigeon pea....

line 69: use full farm at first ROS "reactive oxygen species"

Line 116 to 120, if anyone used previous this kit, provide his reference, 

line 130, remove ROS from heading

figure 1, quality very low,

line 194, 19.47-, 3.72- and 2.68- mg/g, respectively.

Figure 2 and 3, figure quality very low

The decimal of value reported in all tables of the manuscript must be uniform as the value of the detection limit of the used method.

Figure 5, quality very low

Figure 6A , quality very low

Line 416: use full farm at first use, SOD,

Line 440-441, rewrite the lines

Conclusion very short

Please correct all grammatical errors and typos

Recheck all abbreviations.

Author Response

Response to reviewer comments and suggestions

Reviewer 3

Comments and Suggestions for Authors

The overall manuscript is written very well. However, before acceptance author should improve these mistakes.

  1. Line 15, rewrite, "commonly known as pigeon pea"

Ans: We have corrected as “English name is pigeon pea” page 1, line 15 highlight in red.

  1. Line 15: what CR?

Ans: CR is the abbreviation of C. cajan (L.) Millsp roots, we have changed in full name page 1, line 15 highlight in red.

  1. Line 28-29, Rewrite concluding lines, clearly brief in two or three lines.

Ans: We have rewritten our concluding clearly brief in two or three lines, page 1, line 27-30, highlight in red.

  1. Line 43, same line write in abstract, remove and start sentence from pigeon pea....

Ans: We have corrected as follows: “C. cajan (L.) Millsp., known as pigeon pea, is a highly valued medicinal plant and cultivated in the tropics and the semi-arid tropics.” page 1, line 44-45 highlight in red.

  1. Line 69: use full farm at first ROS "reactive oxygen species"

Ans: We have replaced ROS as "reactive oxygen species" page 2, line 61 highlight in red.

  1. Line 116 to 120, if anyone used previous this kit, provide his reference,

Ans: Thank you for your valuable comments. We have added the reference [31] (Song et al in methods page 4, line 130-131 highlight in red).

  1. Line 130, remove ROS from heading

Ans: We have deleted ROS from heading

  1. Figure 1, quality very low,

Ans: We have improved quality of Figure 2 in page 6

  1. Line 194, 19.47-, 3.72- and 2.68- mg/g, respectively.

Ans: We have added 19.47-, 3.72- and 2.68- mg/g, respectively (Page 7, line 199 highlight in red).

  1. Figure 2 and 3, figure quality very low

Ans: We have improved quality of Figure 3 and 4 page 7-8.

  1. The decimal of value reported in all tables of the manuscript must be uniform as the value of the detection limit of the used method.

Ans: Thank you for your valuable comment. We have corrected all tables.

  1. Figure 5, quality very low

Ans: We have improved quality of Figure 6 page 11

  1. Figure 6A, quality very low

Ans: We have improved quality of Figure 7A page 12

  1. Line 416: use full farm at first use, SOD,

Ans: We changed SOD as superoxidase dismutase page 17, line 402 highlight in red.

  1. Line 440-441, rewrite the lines

Ans: The sentence was corrected as “Thus, EECRp70 induced cell apoptosis and inhibit cell proliferation was related with the induction of cell cycle arrest.” page 17, line 427-428 highlight in red.

  1. Conclusion very short

Ans: We have added contents in conclusion, page 20, line 515-518.

  1. Please correct all grammatical errors and typos

Ans: We have corrected all grammatical errors and typos.

  1. Recheck all abbreviations.

Ans: Abbreviations have been corrected.

Reviewer 4 Report

- I didn't understand why the authors highlight in red some paragraphs.

- Figures need to be improved.

- The time of incubation could be extended for synergy essay to more than 24h.

- Synergy of antiproliferation was performed between EECRp70 and Cisplatin or Taxol 2. But not with a combination of EECRp70 with Genistein. Why?

- Deepen investigation are needed to determine the safety threshold of EECRp70 as a potentiel adjuvant for chemotherapy.

Author Response

Response to reviewer comments and suggestions

Reviewer 4

Comments and Suggestions for Authors

  1. I didn't understand why the authors highlight in red some paragraphs.

Ans: Some paragraphs were highlight in red is due to response to academic editor questions.

  1. Figures need to be improved.

Ans: All figures have been improved.

  1. The time of incubation could be extended for synergy essay to more than 24h.

Ans: We have done synergistic effects with different incubation times of 12, 24, and 48 h, but the results of synergistic effect of 24 h were the most significant, so the results of this incubation time (24 h) are presented.

  1. Synergy of antiproliferation was performed between EECRp70 and Cisplatin or Taxol 2. But not with a combination of EECRp70 with Genistein. Why?

Ans: We expect that EECRp70 can be used in adjuvant chemotherapy with chemotherapy drugs in the future, so EECRp70 was selected for synergistic therapy with taxol and cisplatin.

  1. Deepen investigation are needed to determine the safety threshold of EECRp70 as a potential adjuvant for chemotherapy.

Ans: Thanks for valuable comments. As far there is no published article on the safe dose of EECRp70, but we recently reported in a rat study that the NOAEL for EECR is approximately 1 g/kg bw [63] which by extrapolation from rats to humans, is equivalent to approximately 972 mg/60 kg person/day [24]. We have added in discussion section page 19, line 499-501 highlight in red.

[63] Song, T.Y. The development of the teeth‑protecting ingredients of Cajanus cajan (L.) Millsp. Root. In: Final Report of the Agriculture and Food Administration of the Agriculture Committee of the Executive Yuan

(110AS‑1.6.1‑FD‑Z5), Taiwan: Committee of Agriculture, 2021, pp. 1‑30.

[24] Yang, S.E.; Lin, Y.F.; Liao, J.W.; Chen, J.T.; Chen, C.L.; Chen, C.I.; Hsu, S.L.; Song, T.Y. Insulin Sensitizer and Anti-hyperlipidemic Effects of Cajanus cajan (L.) Millsp. Root in Methylglyoxal‑Induced Diabetic Rats. Chin. J. Physiol. 2022, 65, 125-135.

Round 2

Reviewer 2 Report

No doubt authors have improved the quality of manuscript. But the quality of the figures is not good. Few are over stretched and others are blurry. I am not sure how authors will improve it but it is highly needed for publication in "Agriculture". 

Author Response

Dear reviewer,

We have improved the quality of all figures in the Manuscript.

Expect all graphs to meet your standards.

Best Regards,

Tuzz-Ying Song 

Reviewer 3 Report

accepted

Author Response

Thank's for your kind helps! 

Your valuable comments have greatly improved the quality of the manuscript.